# Human-induced pluripotent stem cell-derived microglia integrate into mouse retina and recapitulate features of endogenous microglia

Wenxin Ma[1], Lian Zhao[2], Biying Xu[3], Robert N Fariss[4], T Michael Redmond[5], Jizhong Zou[6], Wai T Wong[7]*, Wei Li[1]*

[1]Retinal Neurophysiology Section, National Eye Institute, Bethesda, United States; [2]Genetic Engineering Core, National Eye Institute, Bethesda, United States; [3]Immunoregulation Section, National Eye Institute, Bethesda, United States; [4]Biological Imaging Core, National Eye Institute, Bethesda, United States; [5]Molecular Mechanisms Section, National Eye Institute, Bethesda, United States; [6]iPSC Core, National Heart, Lung, and Blood Institute, Bethesda, United States; [7]Tiresias Bio, Half Moon Bay, United States

*For correspondence:
Wai@tiresiasbio.com (WTW);
liwei2@nei.nih.gov (WL)

Competing interest: The authors declare that no competing interests exist.

## eLife assessment

The authors have improved a method to differentiate human iPSC-derived microglial cells with immune responses and phagocytic abilities; and through transplantation into the adult mouse retina, the authors further demonstrated their integration and occupation of native microglial cell space, and functional response to retinal injuries. The study is **important** and the data are **convincing** for potential microglial replacement therapy to treat retinal and CNS diseases.

**Abstract** Microglia exhibit both maladaptive and adaptive roles in the pathogenesis of neurodegenerative diseases and have emerged as a cellular target for central nervous system (CNS) disorders, including those affecting the retina. Replacing maladaptive microglia, such as those impacted by aging or over-activation, with exogenous microglia that can enable adaptive functions has been proposed as a potential therapeutic strategy for neurodegenerative diseases. To investigate microglia replacement as an approach for retinal diseases, we first employed a protocol to efficiently generate human-induced pluripotent stem cell (hiPSC)-derived microglia in quantities sufficient for in vivo transplantation. These cells demonstrated expression of microglia-enriched genes and showed typical microglial functions such as LPS-induced responses and phagocytosis. We then performed xenotransplantation of these hiPSC-derived microglia into the subretinal space of adult mice whose endogenous retinal microglia have been pharmacologically depleted. Long-term analysis post-transplantation demonstrated that transplanted hiPSC-derived microglia successfully integrated into the neuroretina as ramified cells, occupying positions previously filled by the endogenous microglia and expressed microglia homeostatic markers such as P2ry12 and Tmem119. Furthermore, these cells were found juxtaposed alongside residual endogenous murine microglia for up to 8 months in the retina, indicating their ability to establish a stable homeostatic state in vivo. Following retinal pigment epithelial cell injury, transplanted microglia demonstrated responses typical of endogenous microglia, including migration, proliferation, and phagocytosis. Our findings indicate the feasibility of microglial transplantation and integration in the retina and suggest that modulating microglia through replacement may be a therapeutic strategy for treating neurodegenerative retinal diseases.

## Introduction

Microglia are the innate immune cells of the central nervous system (CNS), including the retina, and play pivotal roles in neuronal (*Puñal et al., 2019*; *Anderson et al., 2019*; *Huang et al., 2012*) and vascular development (*Checchin et al., 2006*; *Ritter et al., 2006*; *Kubota et al., 2009*), normal synapse formation (*Stevens et al., 2007*; *Schafer et al., 2012*; *Schafer and Stevens, 2015*; *Hong et al., 2016*), maintaining local homeostasis in the neural environment (*Li and Barres, 2018*; *Colonna and Butovsky, 2017*; *Sierra et al., 2013*), and the regulation of immune activity (*Okunuki et al., 2019*). Conversely, they are also implicated in driving pathologic progression in various retinal diseases, including age-related macular degeneration (AMD) (*Combadière et al., 2007*; *Ma et al., 2009, Ma et al., 2012*; *Karlstetter et al., 2015*), glaucoma (*Bosco et al., 2019*; *Ramírez et al., 2020*), diabetic retinopathy (*Xu and Chen, 2017*), and uveitis (*Broderick et al., 2002*; *Okunuki et al., 2019*; *Zhou et al., 2020*).

Under homeostatic conditions in the adult retina, microglial cells are predominantly distributed in the inner plexiform layer (IPL) and outer plexiform layer (OPL) and vigilantly survey environmental changes through dynamic surveying behavior in their ramified processes (*Lee et al., 2008*). Their presence and homeostatic function are crucial for maintaining normal retinal functions, including the maintenance of synaptic function and integrity (*Wang et al., 2016*). Under normal conditions, microglial cells sustain equilibrium in their endogenous numbers via slow self-renewal (*Réu et al., 2017*). However, with the onset of pathology, this homeostasis can be disrupted following microglia activation, migration, and proliferation. Microglia repopulation in the retina following perturbation is achieved through both the proliferation of endogenous microglia and the infiltration of peripheral monocytes (*Ma et al., 2017*; *Huang et al., 2018*; *Zhang et al., 2018*).

Studies of microglia cell repopulation have indicated that retinal resident microglia can sustain their population with the local microglial cell dividing and migration if any perturbations do not exceed the threshold of the recovery speed by local neighbor microglia. However, in cases of more severe retinal injury or infection that cause significant redistribution of endogenous microglia, the reestablishment of retinal microglial homeostasis will, in addition, involve peripheral monocytes that infiltrate into the retina to take up residence as macrophages. The ability of the retina to incorporate exogenous monocytic cells suggests that microglia cell replacement employing exogenously introduced microglia may be feasible and can exert therapeutic effects post-injury. Inhibiting retinal microglia over-activation has shown efficacy in animal models of retinal injury (*Zhao et al., 2011*; *Karlstetter et al., 2015*; *Au and Ma, 2022*) and potential signal in early-phase clinical trials (*Cukras et al., 2012*). These observations suggest that the depletion of maladaptive microglia in pathological contexts and their replacement with microglia that have a more homeostatic phenotype may constitute a potential therapeutic strategy.

Studies of microglia have largely been performed in rodent-derived models, largely due to the accessibility of various transgenic disease models. However, several studies have indicated that genetic and functional differences exist between murine and human microglia (*Dawson et al., 2018*; *Friedman et al., 2018*; *Ueda et al., 2016*). For instance, microglia-expressed genes CD33 and CR1, which have been associated with Alzheimer's disease (AD) risk in genome-wide association studies, lack reliable orthologs in mice (*Hasselmann and Blurton-Jones, 2020*). Additionally, over half of the AD risk genes that are enriched in microglia demonstrate <70% sequence homology between humans and mice (*Hasselmann and Blurton-Jones, 2020*). There are also significant differences in the levels of protein expression of some microglia-expressed complement factors and inflammatory cytokines between humans and mice (*Galatro et al., 2017*; *Gosselin et al., 2017*; *Smith and Dragunow, 2014*). As a result, microglia from murine models may not accurately represent those found in human conditions (*Burns et al., 2015*), limiting their translational potential.

To study microglia of human origin, some investigators have attempted to isolate microglial cells from human tissue. However, owing to limitations in sample availability, and the rapid transcriptomic changes that ex vivo microglia undergo post-isolation, these studies have been technically constrained (*Bohlen et al., 2017*; *Butovsky et al., 2014*; *Gosselin et al., 2017*). As an alternative to primary microglia, human-induced pluripotent stem cells (iPSCs) offer a wealth of possibilities and have been increasingly used to generate human microglia via differentiation in vitro (*Muffat et al., 2016*; *Pandya et al., 2017*; *Abud et al., 2017*; *Douvaras et al., 2017*; *Haenseler et al., 2017*; *Takata et al., 2017*). This approach has enabled the generation of a large quantity of cells of a specified

genetic background, enabling the creation of in vivo human microglia cell models through xenotransplantation of human-induced pluripotent stem cell (hiPSC)-derived microglia into the murine CNS (*Abud et al., 2017*; *Svoboda et al., 2019*; *Parajuli et al., 2021*; *Xu et al., 2020a*; *Chadarevian et al., 2023*).

In this study, we adopted a previously published protocol appropriate for culturing hiPSC-derived microglial cells (*Muffat et al., 2016*; *Pandya et al., 2017*; *Abud et al., 2017*; *Douvaras et al., 2017*; *Haenseler et al., 2017*; *Takata et al., 2017*). We characterized microglia differentiation by examining the RNA and protein expression levels of microglia-enriched genes. We also assessed the inflammatory responses and phagocytic functions of hiPSC-derived microglia in vitro. We then established a human iPSC-derived microglia cell model through the xenotransplantation of human iPSC-derived microglial cells into the retina of an adult mouse. We found that when transplanted by subretinal injection, human iPSC-derived microglial cells were able to migrate into the retina where native retinal microglia reside and acquire a morphology resembling endogenous mouse microglia, and express microglia signature markers. These grafted cells persisted in the mouse retina for at least 8 months and responded to retinal pigment epithelial (RPE) cell injury in ways resembling endogenous mouse microglia. Xenografting of hiPSC-derived microglia into mouse retina has the promise of being used to create in vivo models of retinal disease and injury to evaluate the preclinical efficacy of potential therapeutic agents, as well as to evaluate microglia transplantation itself as a potential therapeutic intervention.

## Results
### Differentiation and characterization of human iPSC-derived microglia

We used five distinct human iPSC lines for microglia cell differentiation, including the first-available iPSC line, KYOUDXR0109B, from ATCC; and four other lines NCRM6, MS19-ES-H, ND2-AAVS1-iCAG-tdTomato, and NCRM5-AAVS1-CAG-EGFP, from the National Heart, Lung, and Blood Institute (NHLBI). Our approach to microglia differentiation was informed by our previous work with primary mouse retinal microglia cell culture (*Ma et al., 2009*) and a variety of established microglia cell differentiation protocols (*Muffat et al., 2016*; *Pandya et al., 2017*; *Abud et al., 2017*; *Douvaras et al., 2017*; *Haenseler et al., 2017*; *Takata et al., 2017*). We opted for the myeloid progenitor/microglia cell floating culture method (*van Wilgenburg et al., 2013*; *Haenseler et al., 2017*) for its simplicity, efficiency, and consistency, which enables the generation of a large and uniform population of microglial cells.

The differentiation process involved three key stages: embryoid body (EB) formation, myeloid progenitor cell generation, and microglia cell maturation (*Figure 1A–D*). Following myeloid differentiation, floating myeloid progenitor cells were harvested and allowed to differentiate further for 2 weeks in 6-well plates/flask under conditions promoting microglial differentiation. We modified this step to include additional differentiation factors in the differentiation medium, including IL34, CSF1, CX3CL1, TGFB1, and TGFB2. We found that these factors promoted microglial morphological ramification and process elongation. Immunohistochemical analysis of the resulting cells showed that among CD34(+) cells, 98.6% were immunopositive for IBA1 (*Figure 1E, F*), and 98.5% were immunopositive for P2RY12 (*Figure 1E, G*). Immunostaining with myeloid cell markers CX3CR1, CD68, and CD11b showed positivity in 88%, 99.7%, and 94.3%, respectively, demonstrating the high efficiency of differentiation achieved by this procedure (*Figure 1—figure supplement 1*). Most of the resulting hiPSC-derived microglia showed spindle-shaped morphologies, with some displaying short ramifications in their processes (*Figure 1E*), resembling those observed in primary mouse retinal microglia cultures (*Ma et al., 2009*). Floating myeloid progenitor cell harvest could be performed repeatedly over 3 months following culture establishment, providing a steady and consistent supply for further microglia differentiation and generation.

A comparative RNAseq analysis between differentiated hiPSC-derived microglia collected at the end of the protocol vs. floating myeloid progenitor cells revealed a significant upregulation of microglia-enriched gene expression following microglial differentiation, including *Cx3cr1*, *P2ry12*, *P2ry13*, *Aif1*, *Trem2*, *Gpr34*, *CD53*, *CTSS*, and *C3aR1* (*Figure 2A–C*). Moreover, hiPSC-derived microglia exhibited higher expression of genes associated with inflammation, apoptosis regulation, phagocytosis, lipid metabolism, and immune responses. The floating myeloid progenitor cells showed higher expression

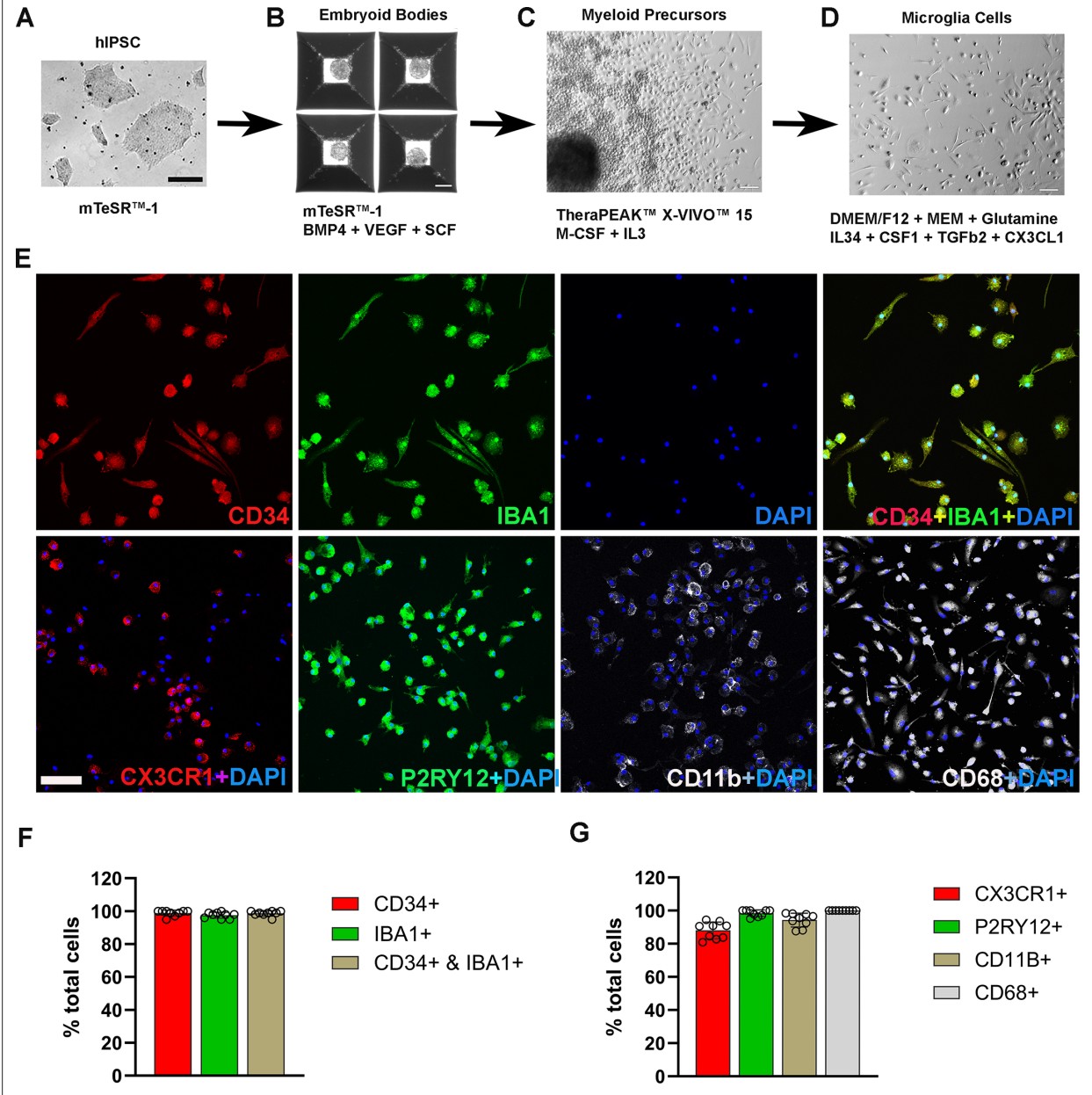

**Figure 1.** Differentiation and characterization of human-induced pluripotent stem cell (iPSC)-derived microglia. (**A**) Human iPSCs were cultured in a 6-well plate. Scale bar = 200 μm. (**B**) Embryoid body formation was enabled in AggreWell800 plate at day 8 in culture medium mTeSR1 plus BMP4, VEGF, and SCF. Scale bar = 200 μm. (**C**) Image of a myeloid precursor cluster following 1 month culture of embryoid bodies in TheraPEAK X-vivo-15 Serum-free Hematopoietic Cell Medium with added M-CSF and IL3. Scale bar = 50 μm. (**D**) Image of microglial cells in maturation culture for 2 weeks with Dulbecco's Modified Eagle Medium (DMEM)/F12 plus non-essential amino acids, glutamine, IL34, CSF1, TGFb2, and CX3CL1. Scale bar = 50 μm. (**E**) Immunohistochemical staining for *Iba1* and human CD34, CX3CR1, P2RY12, CD11b, and CD68. Scale bar = 100 μm. (**F**) Cell counts and colocalization analysis of (**F**) CD34- and Iba1-positive cells and (**G**) positivity for myeloid cell markers CX3CR1, CD11b, activation marker CD68, and microglia marker P2RY12 in differentiated microglia.

The online version of this article includes the following figure supplement(s) for figure 1:

**Figure supplement 1.** Immunocytochemistry staining with human SPI1 and TREM2.

of hematopoietic/myeloid cell lineage genes (*Figure 2D*). Ingenuity Pathway Analysis (IPA) of differentially expressed genes identified IL6, IL1B, and STAT3 as central signaling hubs critical to the regulation of inflammatory responses in microglia (*Figure 2E*, *Figure 2—figure supplement 1*, *Supplementary file 1*). We also compared the gene expression profiles between differentiated hiPSC-derived

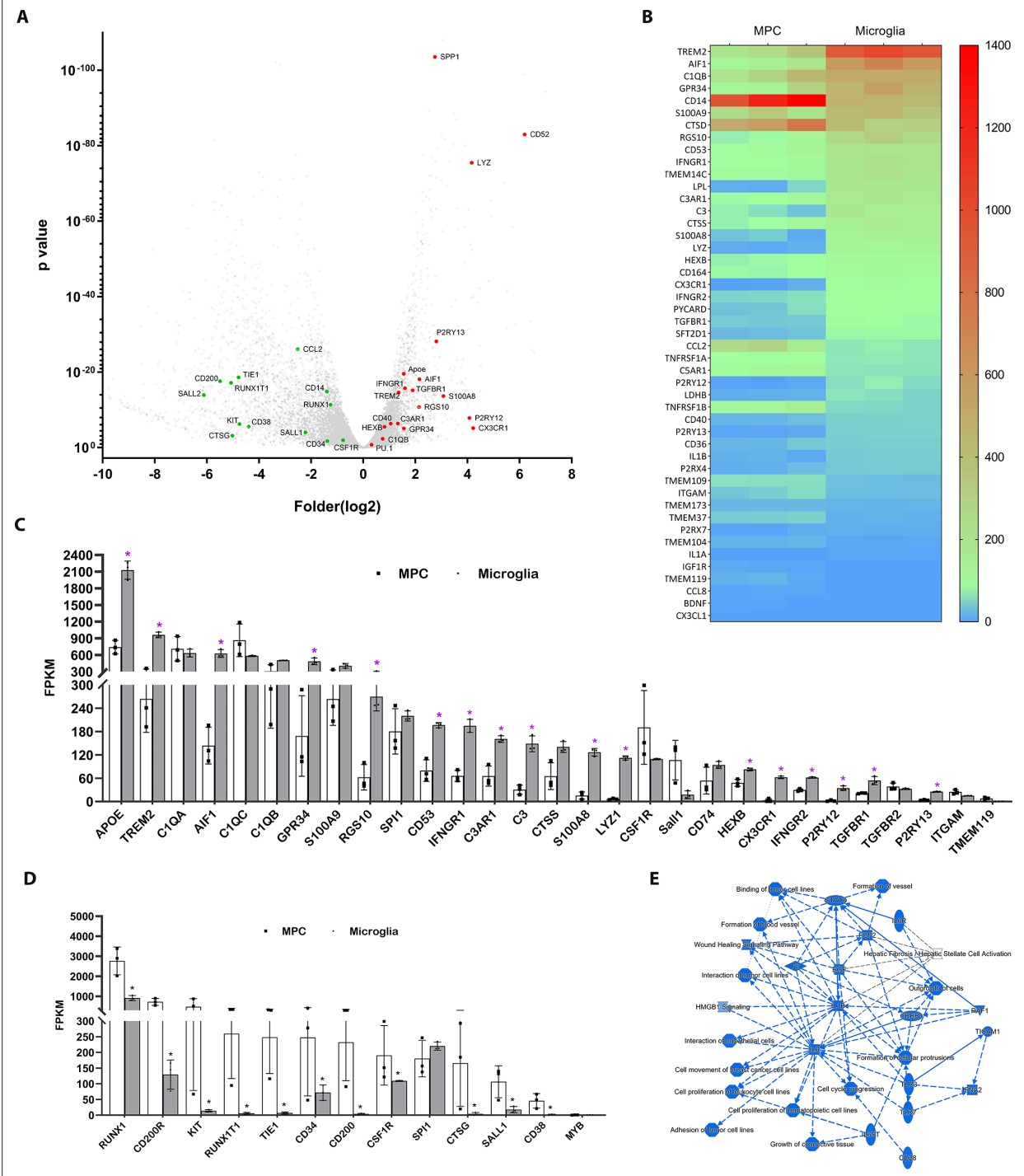

**Figure 2.** Profiling of genes differentially expression between differentiated microglial cells vs. myeloid progenitor cells (MPCs) using bulk RNAseq analysis. (**A**) Volcano plot showing representative genes that were either upregulated (red) or downregulated (green) in differentiated microglia vs. MPCs. (**B**) Heat map showing increased expression of microglia-enriched genes in differentiated microglia (*Supplementary file 1*). (**C**) Histogram comparing the expression levels of microglia-enriched genes in terms of Fragments Per Kilobase of transcript per Million mapped reads (FPKM). *p < 0.05. (**D**) Histogram comparing expression levels of myeloid cell lineage genes in human-induced pluripotent stem cell (iPSC)-derived MPC and microglia cells using FPKM. *p < 0.05. (**E**) Graphic signaling pathway analysis with Ingenuity Pathway Analysis (IPA) highlighting *IL6* and *IL1B* as signaling hubs in differential gene expression patterns (*Supplementary file 1*) in differentiated microglia vs. MPCs.

The online version of this article includes the following figure supplement(s) for figure 2:

*Figure 2 continued on next page*

*Figure 2 continued*

**Figure supplement 1.** Analysis results of the canonical pathway of complete differentiated human-induced pluripotent stem cell (hiPSC)-derived microglia vs. myeloid progenitor cells with Ingenuity Pathway Analysis (IPA) with the enriched microglia genes (*Supplementary file 1*).

**Figure supplement 2.** Hierarchical cluster analysis on microglia-enriched genes among human-induced pluripotent stem cell (hiPSC)-derived microglial cells (hiPSC-MG) and human adult brain microglia cells (AMG), fetal brain microglia cells (FMG), inflammatory monocytes (IM), monocytes (M).

**Figure supplement 3.** Correlation analysis between the expression levels of microglia-enriched genes in human-induced pluripotent stem cell (hiPSC)-derived microglia cells (hiPSC-MG) vs. those in other myeloid cells.

**Figure supplement 4.** Correlation analysis between the expression levels of genes in human-induced pluripotent stem cell (hiPSC)-derived microglia cells (hiPSC-MG) vs. human donor brain microglia by sex.

microglial cells vs. human microglia isolated ex vivo from the fetal and adult brain (*Figure 2—figure supplements 2–4 Supplementary files 2 and 3*; *Abud et al., 2017*; *Douvaras et al., 2017*; *Muffat et al., 2016*; *Böttcher et al., 2019*; *van der Poel et al., 2019*; *Eisenberg and Levanon, 2013*). The results of the correlation analysis indicated that hiPSC-derived microglial cells demonstrated an expression profile comparable to those in fetal and adult microglia in vivo but were more distinct from those in monocytes and inflammatory monocytes. Thus, our method of obtaining differentiated microglia can reliably and efficiently generate a large population of homogenous functional microglial cells of human origin.

## Human iPSC-derived microglia show inflammation responses and phagocytosis activity

Microglia play crucial roles in mediating inflammatory responses to stimuli and in phagocytosing pathogens. To investigate these functions further, we stimulated hiPSC-derived microglia with lipo-polysaccharide (LPS) and analyzed their responses. Transcriptomic profiling using bulk RNAseq revealed that the primary responses to LPS stimulation involved changes in *IL6*, *IL1B*, *IL1A*, *TNFA*, and *IFNG* signaling (*Figure 3A*, *Supplementary file 4*), indicating the ability of hiPSC-derived microglia to demonstrate classical activation. This was confirmed through expression analysis using quantitative reverse transcription-PCR (qRT-PCR) and protein multiplex profiling, which showed a 50- to 800-fold increase in the expression of *IL6*, *IL1A*, *IL1B*, *TNFA*, *IL8*, *CXCL10*, and *CCL2* mRNA after 6 hr of LPS stimulation (*Figure 3B*). Similarly, we observed a significant increase in protein expression levels of these cytokines in cell lysate and culture medium (*Figure 3C, D*). We also treated hiPSC-derived microglia with IFNγ and a combination of IFNγ + LPS (*Figure 3—figure supplement 1*), and the results demonstrated that the combination of IFNγ + LPS promoted *IL1A*, *IL1B*, *TNFA*, *CCL8*, and *CXCL10* expression. These findings indicate that hiPSC-derived microglia, akin to microglia in vivo, exhibited strong responses to LPS and IFNγ stimulation.

Microglia are local immune cells in the CNS, functioning as phagocytes involved in clearing apoptotic or necrotic cells, and cell debris (*Green et al., 2016*), remodeling neuronal connectivity by engulfing synapses, axonal and myelin debris (*Paolicelli et al., 2011*), and removing pathogens by direct phagocytosis (*Nau et al., 2014*). To assess the phagocytic capability of hiPSC-derived microglia, we exposed them to three different types of bioparticles: *Escherichia coli* bacteria, zymosan, and bovine photoreceptor outer segments (POSs) (*Figure 4A–D*). The cells altered their morphology in response and rapidly internalized the fluorescent-labeled particles (*Figure 4A–D*, *Figure 4—figure supplements 1 and 2*). The engulfed bioparticles were condensed into perinuclear aggregates, likely within lysosomal bodies. They also demonstrated concurrent morphological changes into amoeboid-shaped cells (*Figure 4D, E*), resembling phenotypes demonstrated by retinal microglia phagocytosing photoreceptors in the context of photoreceptor degenerative pathologies in vivo (*Zhao et al., 2015*).

## Transition of hiPSC-derived microglial cells to a homeostatic state within the mouse retina following xenotransplantation in vivo

To assess the ability of hiPSC-derived microglia to serve as microglia donor cell sources for transplantation, we conducted xenotransplantation experiments using humanized immunodeficient Rag2$^{-/-}$;IL-2rg$^{-/-}$;hCSF1$^{+/+}$ transgenic mice as recipients as previously employed (*Svoboda et al., 2019*; *Xu et al., 2020b*; *Chadarevian et al., 2023*). Prior to xenotransplantation, recipient transgenic mice were pharmacologically depleted of endogenous retinal microglia by systemic administration of the

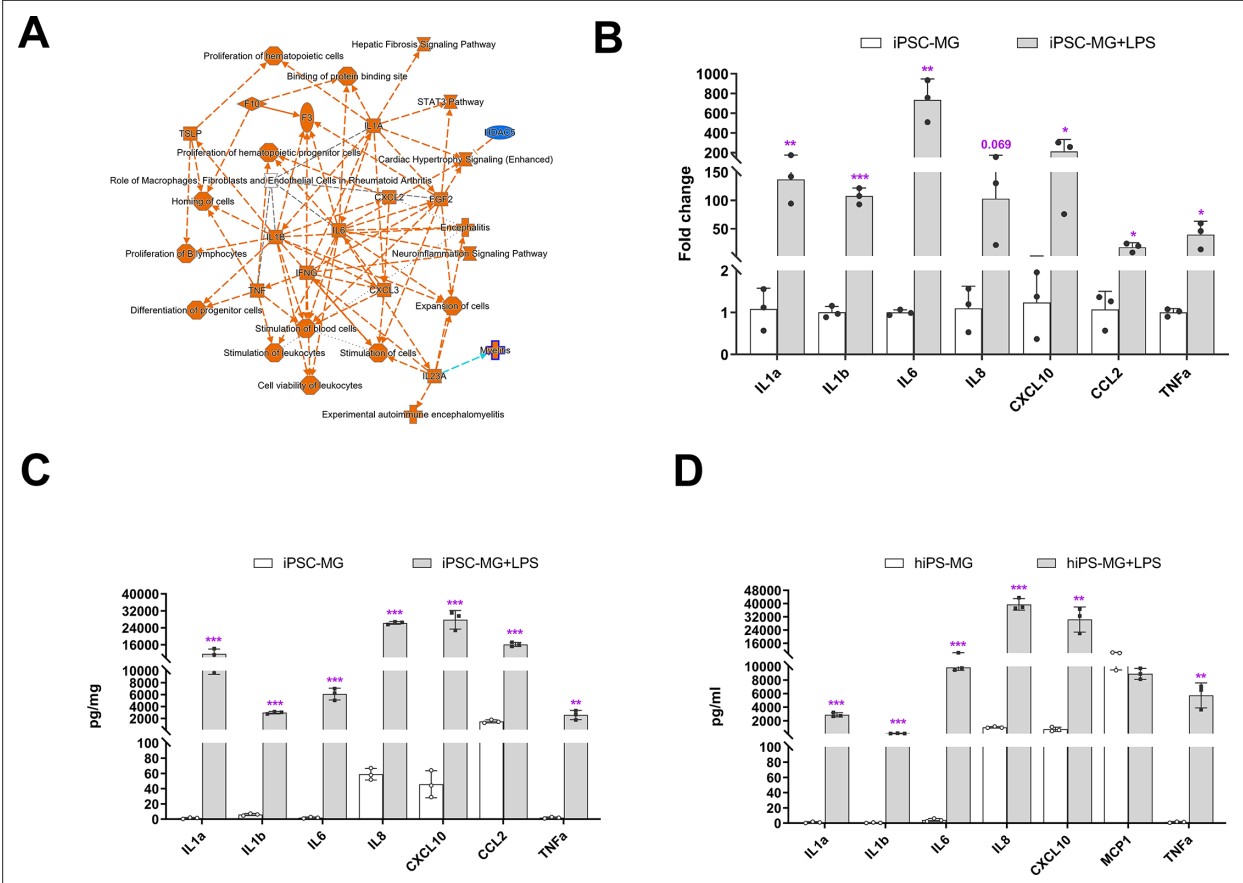

**Figure 3.** Inflammation responses of human-induced pluripotent stem cell (hiPSC)-derived microglial cells following lipopolysaccharide (LPS) stimulation. (**A**) Ingenuity Pathway Analysis (IPA) showed different gene expression (fold change >twofold, p < 0.05) between LPS-treated and control hiPSC-derived microglial cells, demonstrating activation of core pathways involving *IL6*, *IL1A*, *IL1B*, and *IFNG*. (**B**) Assessment of mRNA expression of selected genes for inflammatory cytokines using quantitative reverse transcription-PCR (qRT-PCR; Oligonucleotide primers are provided in Supplementary 4) demonstrated increased expression following LPS (0.1 µg/ml) stimulation for 6 hr (3-6 replicates). These changes corresponded to increases in the protein expression levels of inflammatory cytokines following 24 hr of LPS stimulation as measured with a Multiplex kit (Millipore) in cell lysate (**C**) and conditioned media (**D**). The data in (**C**) and (**D**) are presented as means ± SEM (3-6 replicates). *p < 0.05, **p < 0.01, ***p < 0.001.

The online version of this article includes the following figure supplement(s) for figure 3:

**Figure supplement 1.** Inflammatory cytokines were produced synthetically by IFNG and LPS in human-induced pluripotent stem cell (hiPSC)-derived microglial cells.

CSF1R inhibitor PLX-5622 to create a depleted tissue niche for the integration of xenotransplanted microglia (*Zhang et al., 2018*). Two days following PLX-5622 treatment, adult transgenic mice were transplanted with 5000 hiPSC-derived microglia via injection into the subretinal space (*Figure 5A*). Tissue analysis at 4 and 8 months post-transplantation revealed that transplanted cells, which were marked by either tdTomato or EGFP expression, had migrated anteriorly from the subretinal space into the neural retina and were distributed across a wide retinal area within the inner and outer retinal layers including the ganglion cell layer (GCL), IPL, and OPL (*Figure 5B–D*) in retinal loci typically occupied by endogenous microglia. Transplanted cells were immunopositive for IBA1, and human CD11b (hCD11b) (*Figure 5H, I*), as well as for microglia signature markers hP2RY12 and hTMEM119 (*Figure 5D*). Interestingly, the transplanted cells within the retina showed a ramified morphology typical of homeostatic microglia and demonstrated a regularly tiled 'mosaic' distribution in their soma positions. These integrated cells were juxtaposed alongside residual endogenous murine microglia, which showed similar morphology and distribution as in endogenous conditions. This indicated that transplanted hiPSC-derived microglia responded to similar intraretinal cues and inter-microglia neighbor–neighbor interactions that guide the spatial and morphological organization of retinal

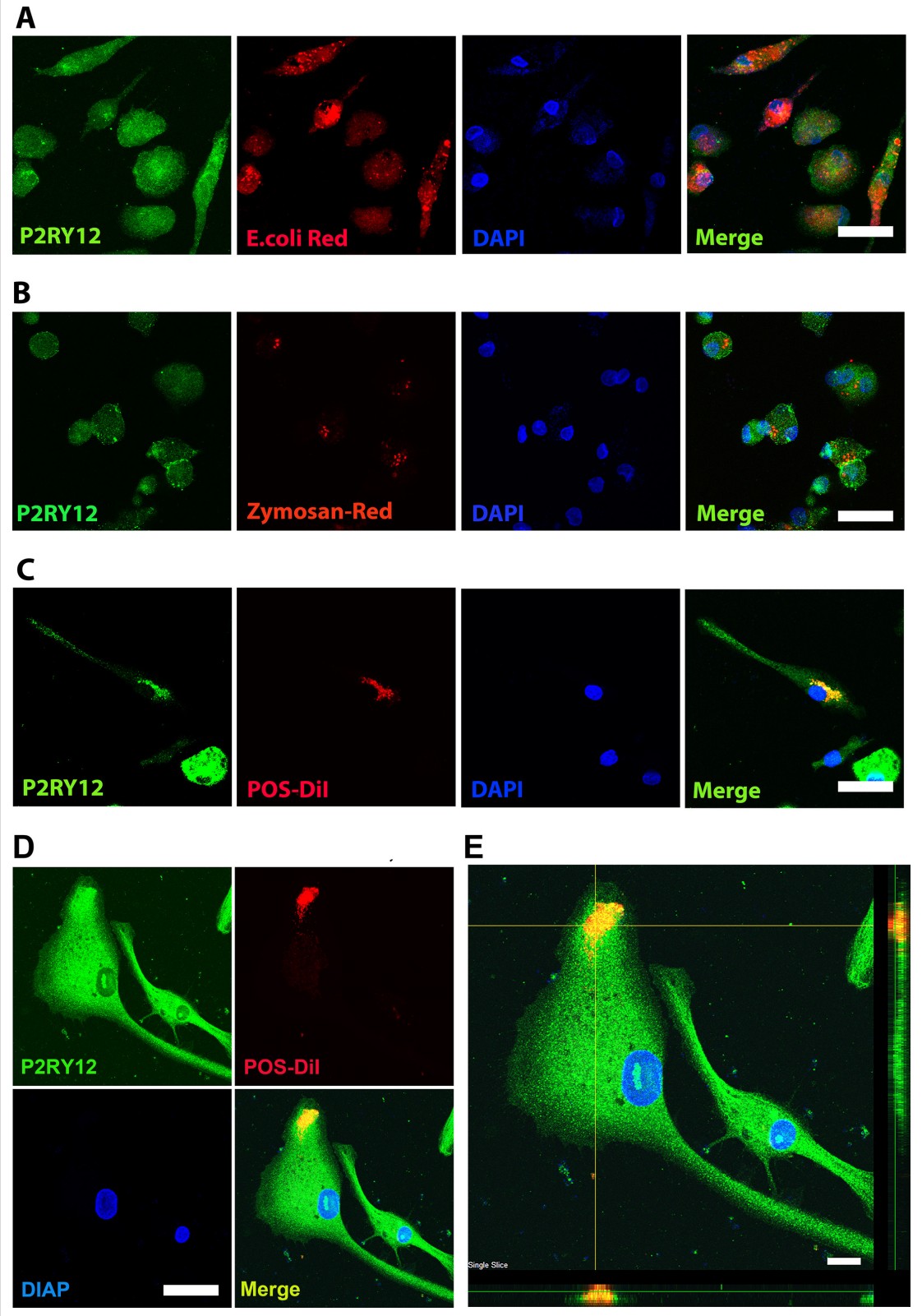

**Figure 4.** Human-induced pluripotent stem cell (iPSC)-derived microglia demonstrate robust phagocytosis. Human iPSC-derived microglia were incubated for 1 hr in pHrodo Red *E. coli* bioparticles (**A**), pHrodo Red zymosan bioparticles (**B**), DiI-labeled bovine photoreceptor outer segments (POSs) (**C**) and labeled with anti-human *P2RY12* antibody (green) and DAPI. Scale bar = 40 µm. (**D**) A high-magnification view of a POS-containing intracellular vesicle within a labeled microglial cell is shown. Scale bar = 40 µm. (**E**) An overlay of panels in (**D**) with side views. Scale bar = 40µm.

*Figure 4 continued on next page*

*Figure 4 continued*

The online version of this article includes the following figure supplement(s) for figure 4:

**Figure supplement 1.** Floating myeloid progenitor cells were cultured in a 2-well slide chamber overnight, and then the DiI-labeled bovine photoreceptor outer segments were added to the chamber and incubated for 1 hr.

**Figure supplement 2.** The morphology of human-induced pluripotent stem cell (hiPSC)-derived microglia cells under 1 hr zymosan treatment with different concentrations.

microglia in vivo (*Figure 5B–G*). Similar observations were made for separate experiments involving the transplantation of tdtomato- and EGFP-expressing hiPSC-derived microglia (*Figure 5B, C, H, I*).

We further evaluated the impact of human iPSC-derived microglia xenotransplantation on the host mouse retinal cells (*Figure 5—figure supplements 1–6*). We examined the effect of microglia transplantation on endogenous Müller cell morphology and gliosis markers as retinal microglia have been described to interact with Müller glia to regulate overall retinal neuroinflammatory response (*Wang et al., 2014*). Immunostaining with glial fibrillary acidic protein (GFAP) and glutamine synthetase (GS) antibody (*Figure 5—figure supplements 1 and 2*) indicated that the transplanted human iPSC-derived microglial cells did not trigger any upregulation of Müller cell gliosis markers or induced morphological changes or reactive proliferation for up to 4 months post-transplantation. Additionally, the laminar organization of the inner and outer retinal layers remained normal and similar to those in control retinas not subjected to transplantation (*Figure 5—figure supplements 1–4*), indicating that microglia transplantation had no adverse impact on the structural integrity of the retina. Furthermore, immunostaining with the mouse CD11b antibody, which marked the residual population of endogenous mouse microglia showed the spatial juxtaposition of these mouse microglia with transplanted human iPSC-derived microglia in a common mosaic of tiled cells (*Figure 5—figure supplements 5 and 6*), indicating the ability of transplanted hiPSC-derived microglia to integrate with and exchange signals with the residual endogenous microglial population, but not replaced.

## Transplanted hiPSC-derived microglia respond to induction of RPE cell injury with migration and proliferation

To monitor the longer-term consequences of microglia xenotransplantation, we extended analysis and examined the location and morphology of transplanted hiPSC-microglia up to 8 months post-transplantation. Analysis of retinal flat mounts confirmed that the tdTomato+ cells remained appropriately located in the GCL, IPL, and OPL, forming a mosaic-like distribution typical of endogenous microglia (*Figure 6—figure supplements 1 and 2*). These cells maintained expression of hP2RY12 and hTMEM119, markers of homeostatic retinal microglia, indicating their long-term integration within the recipient mouse retina.

To evaluate the function of the transplanted hiPSC-derived microglia in the mouse retina and their ability to respond to injury in vivo, we subjected recipient mice 240 days post-transplantation to sodium iodate (NaIO$_3$)-mediated RPE injury and analyzed retinal tissue 3 and 7 days post-injury (*Figure 6A*). In a previous study (*Ma et al., 2017*), we had characterized the responses of endogenous retinal microglia in this injury model; in the few days following injury, microglia within the neural retina migrated into the subretinal space, coming into close proximity to damaged RPE cells. This resulted in a transient decrease in microglia number in the IPL and OPL, which then recovered partially following the proliferation of the remaining microglia to replenish the depleted areas. We found that transplanted hiPSC-derived microglia demonstrated responses similar to endogenous microglia. Three days after NaIO$_3$ injury, there was an increase in tdTomato+, hP2ry12+, hiPSC-derived microglia in the subretinal space, while their number decreased in the IPL and OPL (*Figure 6B–D, G*), indicating a migration of these cells from the neuroretina to the subretinal space. Some of the remaining tdTomato+ and P2ry12+ cells in the inner retina were positive for the cell-proliferation marker Ki67, indicating active cell division (*Figure 6B–D, F*). The numbers of Ki67+ tdTomato+ microglia peaked at 3 days post-injury and decreased thereafter (*Figure 6B–F*). Seven days post-injury, the numbers of tdTomato+ and P2ry12+ human iPSC-derived microglia increased in the IPL and OPL but decreased in the subretinal space (*Figure 6E, G*); Ki67+ tdTomato+ cell numbers also declined (*Figure 6E, F*). These findings suggest that once the hiPSC-derived microglia had replenished endogenous numbers

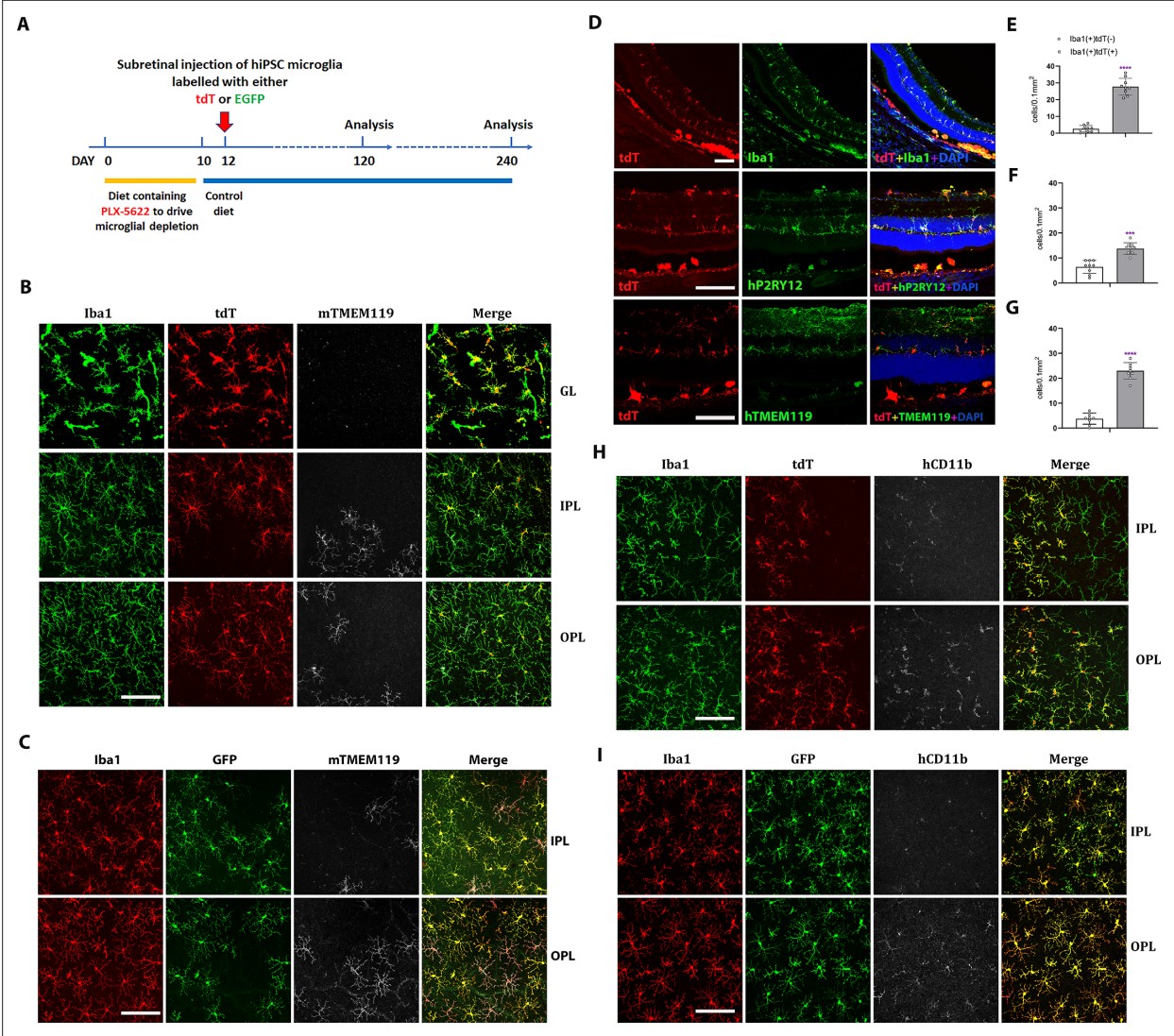

**Figure 5.** Xenotransplanted human-induced pluripotent stem cell (iPSC)-derived microglial cells into recipient mouse retina in vivo demonstrate recapitulation of endogenous distribution, cellular morphology, and stable integration for up to 4 months. (**A**) The schematic diagram shows the timeline for transplantation experiments. Two-month-old adult transgenic Rag2$^{-/-}$;IL2rg$^{-/-}$;hCSF1$^{+/+}$ mice were fed a PLX-5622-containing diet for 10 days before switching to standard chow. Two days following the resumption of standard chow, human iPSC-derived microglial cells expressing either tdTomato or EGFP were xenotransplanted into the subretinal space via subretinal injection (5000 cells in 1 μl injection volume). Retinas were harvested for analysis 120 and 240 days following transplantation. (**B, C**) The retinas isolated from post-transplantation were analyzed in flat-mounted tissue with confocal imaging. Transplanted human-induced pluripotent stem cell (hiPSC)-derived microglia were visualized through their expression of tdtomato (TdT) (**B**) or EGFP (**C**), while endogenous mouse microglia were visualized using immunostaining for mouse Tmem119 (mTmem119). Imaging analysis was performed in separate layers of the retina, including the ganglion cell layer (GL), inner plexiform layer (IPL), and outer plexiform layer (OPL). Scale bar = 100 μm. (**D**) The retinal section showed human iPSC-derived microglial cells integrated into whole retinal layers (top panel) and positively stained with human P2RY12 and TMEM119 microglia signature markers. Scale bar = 100 μm. The microglia cell number in GL, IPL, and OPL of host mouse retina were counted: mouse microglial cells (Iba1+, tdT−) and grafted human microglial cells (Iba1+, tdT+) were shown in (**E**), (**F**), and (**G**), respectively. ***p < 0.001, ****p < 0.0001, 3-6 biological replicates were performed. (**H**) and (**I**) showed tdT (**H**) or EGFP (**I**) labeled human iPSC-derived microglial cells in the IPL and OPL of the flat-mount retina with human CD11b staining. These results demonstrated that the infiltration of grafted hiPSC-derived microglial cells integrated into the mouse retina is general in nature and not cell line specific. Scale bar = 100 μm.

The online version of this article includes the following figure supplement(s) for figure 5:

**Figure supplement 1.** Homeostatic human-induced pluripotent stem cell (hiPSC)-derived microglial cells in the mouse retina do not affect local retinal cells.

**Figure supplement 2.** Homeostatic human-induced pluripotent stem cell (iPSC)-derived microglia cells in the mouse retina do not affect local retinal cells.

*Figure 5 continued on next page*

*Figure 5 continued*

**Figure supplement 3.** Homeostatic human-induced pluripotent stem cell (iPSC)-derived microglia cells in the mouse retina do not affect local retinal cells.

**Figure supplement 4.** Homeostatic human-induced pluripotent stem cell (iPSC)-derived microglia cells in the mouse retina do not affect local retinal cells.

**Figure supplement 5.** Homeostatic human-induced pluripotent stem cell (iPSC)-derived microglia cells in the mouse retina do not take over local retinal microglia cells.

**Figure supplement 6.** Homeostatic human-induced pluripotent stem cell (iPSC)-derived microglia cells in the mouse retina do not take over local retinal microglia cells.

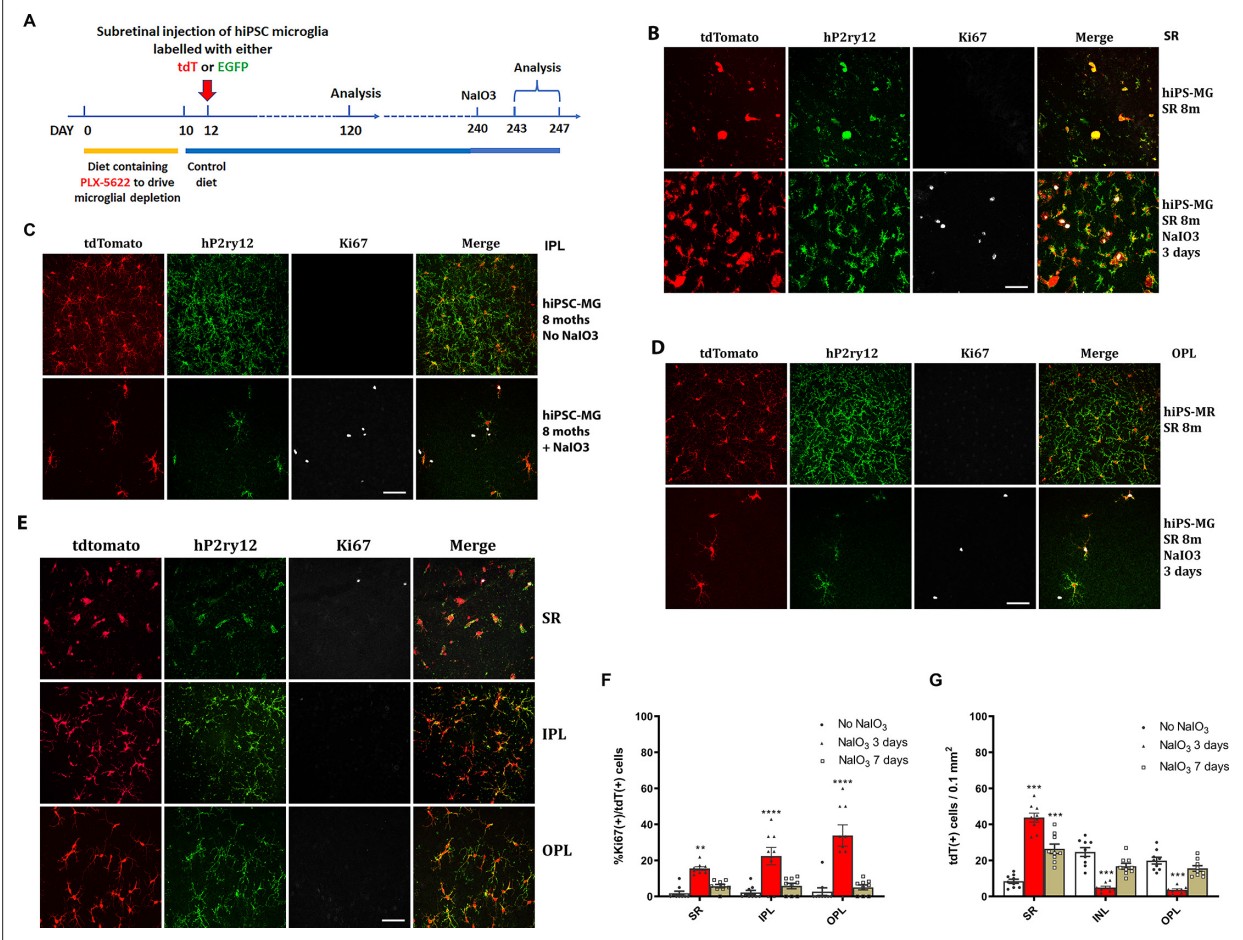

**Figure 6.** Migration and proliferation of human-induced pluripotent stem cell (hiPSC)-derived microglia in the mouse retina after sodium iodate (NaIO₃)-induced retinal pigment epithelial (RPE) cell injury. (**A**) The schematic diagram shows the experiment's procedure. After 8 months post-transplantation of hiPSC-derived microglia, recipient animals were administered NaIO₃ (30 mg/kg body weight, intraperitoneal injection) to induce RPE injury. Retinas were harvested at 3 and 7 days after NaIO₃ administration and microglia numbers in the retina and subretinal space will be monitored in retina and RPE-choroid flat mounts. (**B**) RPE-choroid flat mounts demonstrate an increase of hiPSC-derived microglia (tdTomato+ and P2RY12+) in the subretinal space in response to RPE injury. A subset of subretinal microglia labeled for Ki67 indicates active proliferation. Scale bar = 60 µm. (**C**) and (**D**) showed the number of P2RY12+ and tdtomato+ human microglial cells in inner plexiform layer (IPL) (**C**) and outer plexiform layer (OPL) (**D**) decreased; some of them showed Ki67+ staining, Scale bar = 60 µm. The cell count results showed in (**F**) and (**G**). (**E**) The retinal flat mount showed the number of P2RY12+ and tdtomato+ human microglial cells in IPL and OPL that were repopulated, and the cells stopped dividing with loss of the Ki67 staining at 7 days after NaIO₃ injection. The cell numbers are shown in (**F**) and (**G**) (3 biological replicates). Scale bar = 60 µm, ** P<0.01, *** P<0.001, **** P,0.0001.

The online version of this article includes the following figure supplement(s) for figure 6:

**Figure supplement 1.** The images of hP2RY12 staining on the retina after 8 months of xenotransplantation.

**Figure supplement 2.** The images of hTMEM119 staining on the retina after 8 months of xenotransplantation.

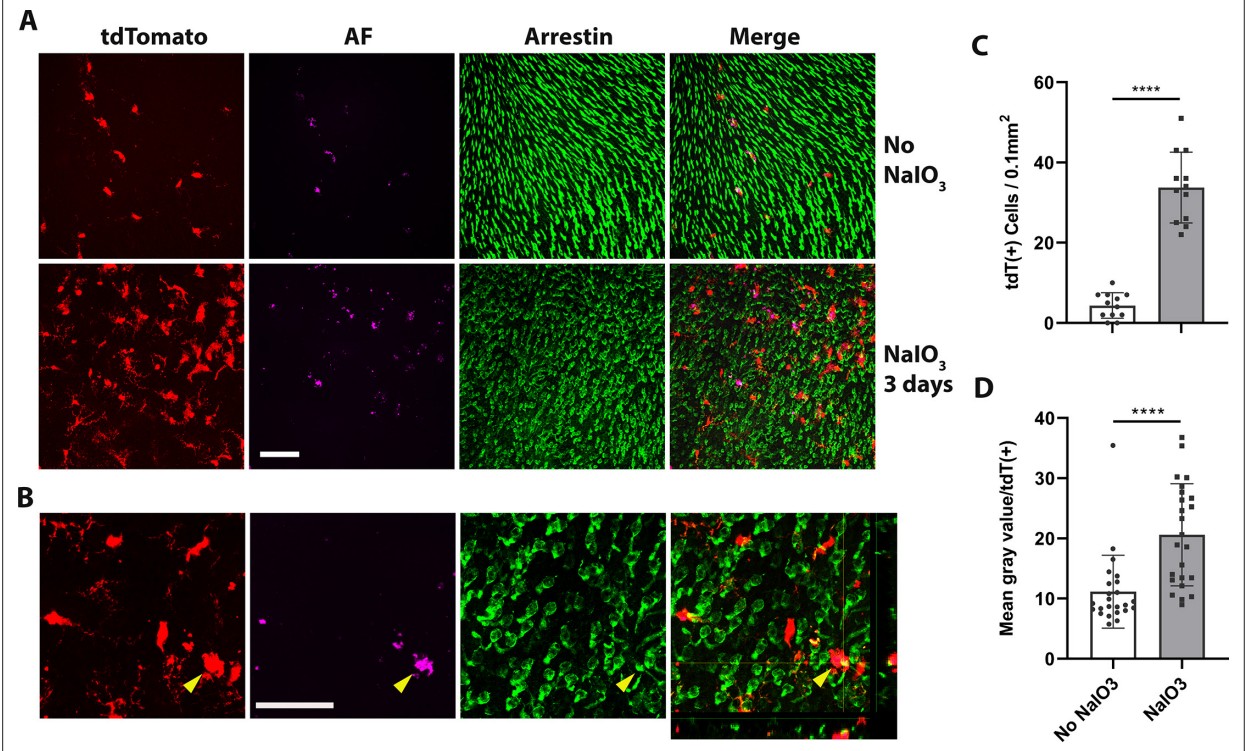

**Figure 7.** Dyshomeostatic human-induced pluripotent stem cell (iPSC)-derived microglial cells in the mouse retina phagocytose dead photoreceptor cells/debris after retinal pigment epithelial (RPE) cell injury. (**A**) Dyshomeostatic human microglial cells (tdtomato+) accumulated in the photoreceptor cell layer after 3 days of sodium iodate ($NaIO_3$)-induced RPE cell injury compared with no $NaIO_3$ administration. The photoreceptor cells stained with cone arrestin (green) and autofluorescence showed in magenta. Scale bar = 60 µm. (**B**) High-magnificent images and the side view showed human microglial cells (red) co-labeled with photoreceptor cells arrestin staining (green) after 3 days of $NaIO_3$ injury. The yellow arrowhead showed the colocalized tdT+ human microglia cell and arrestin+ cone photoreceptor cell. Scale bar = 40 µm. (**C**) The number of tdtomato+ human microglial cells in the photoreceptor layer. (**D**) The mean gray autofluorescence value in each human microglia cell. ****p < 0.0001.

The online version of this article includes the following figure supplement(s) for figure 7:

**Figure supplement 1.** The inflammation, phagocytosis, adhesion and migration, neurotrophic factors, and microglia signature gene expression in human-induced pluripotent stem cell (hiPSC)-derived microglia cells of grafted retinas.

in the inner retina, their division ceased. This response mirrors that of the endogenous mouse retinal microglia to $NaIO_3$ injury (*Ma et al., 2017*).

Overall, these results demonstrate that the transplanted human iPSC-derived microglial cells retained a capacity for migration and proliferative responses to injury in a manner observed for endogenous microglia.

## hiPSC-derived microglial cells phagocytize debris or dead photoreceptor cells after $NaIO_3$-induced RPE cell injury

Phagocytosis, a critical function of microglia, is essential both during development and in the resolution of pathological processes. Retinal microglia adaptively phagocytose and clear apoptotic photoreceptors in the rd10 mouse model of photoreceptor degeneration (*Silverman et al., 2019*). We found here that 3 days following $NaIO_3$-induced RPE cell injury, tdTomato+ transplanted hiPSC-derived microglia migrated not only to the subretinal space but also into the photoreceptor layer (*Figure 7A–C*), coincident with the time of photoreceptor degeneration, when photoreceptor morphology becomes disrupted and photoreceptor density decreases (*Figure 7A*). Within the photoreceptor layer, hiPSC-derived microglia were observed to phagocytose photoreceptors as evidenced by the accumulation of intracellular autofluorescent material in their soma (*Figure 7A, B, D*), and their transformation into larger amoeboid cells (*Figure 7B*, yellow arrowhead) containing arrestin-immunopositive photoreceptor-derived debris. mRNA analysis for human-specific transcripts also

indicated that transplanted hiPSC-derived microglia upregulated inflammatory cytokine expression, increased phagocytosis, and decreased expression of microglia homeostatic genes and neurotrophic factors (*Figure 7—figure supplement 1*; *Supplementary file 5*). Taken together, these findings further demonstrate that the xenografted human iPSC-derived microglial cells generate functional responses to in vivo injury that closely resemble those of endogenous homeostatic retinal microglia.

## Discussion

Microglial cells are instrumental in the development and progression of numerous CNS diseases. For instance, in AD, microglia are enriched in over 50% of associated gene loci implicated in AD risk (*Hasselmann and Blurton-Jones, 2020*). Similarly, in AMD, 57% of 368 genes, located close to 52 AMD gene loci, are expressed in retinal microglial cells (Figure 9), and 52% of them are highly expressed in microglial cells (*Figures 8 and 9*; *Fritsche et al., 2016*; *Ma et al., 2013*; *den Hollander et al., 2022*). Understanding microglia cell functions is essential for investigating disease mechanisms and identifying accurate therapeutic targets. Most of our current knowledge about microglial cells comes from rodent studies. However, genetic and functional differences exist between murine and human microglia (*Galatro et al., 2017*; *Gosselin et al., 2017*; *Smith and Dragunow, 2014*). Hence, more in-depth knowledge of human microglial cells in vitro and in vivo is required.

Human iPSCs offer promising prospects for many retinal research fields (*Zhong et al., 2014*; *Leach et al., 2016*; *Tanaka et al., 2016*). For over a decade, macrophage/microglial cells have been differentiated using human iPSC (*Karlsson et al., 2008*; *Pocock and Piers, 2018*). An abundance of hiPSC-derived microglial cells, with defined genomic background and easy manipulation of hiPSC, offer substantial benefits in various research areas.

Under in vivo physiological conditions, microglial cells exhibit a tile-like arrangement, without overlap. They try to maintain this property in culture, allowing overgrown cells to float out to the medium. Based on this phenomenon and a variety of established microglia cell differentiation protocols (*Muffat et al., 2016*; *Pandya et al., 2017*; *Abud et al., 2017*; *Douvaras et al., 2017*; *Haenseler et al., 2017*; *Takata et al., 2017*), We chose to employ the myeloid progenitor/microglia cell floating culture method (described in *van Wilgenburg et al., 2013*; *Haenseler et al., 2017*) due to its simplicity, efficiency, and consistency. This approach facilitates the generation of a large and uniform population of myeloid progenitor cells, with over 98.6% expressing the *CD34* marker, sustained over a 3-month period. These progenitor cells differentiate into pure microglial cells (>98.5% *P2RY12+*), bearing a profile rich in microglia genes, and demonstrate characteristics similar to native microglia in physiological CNS tissue. A comparison of the signaling pathway to myeloid cells reveals the central hubs of signaling as IL6, IL1b, and the stat3 pathway (*Figure 2D*), which are key to microglia functioning during inflammation. The differentiation protocol employing floating myeloid progenitor cells produces a significant number of CNS resident-like microglial cells. These hiPSC-derived microglial cells respond robustly to LPS stimulation and demonstrate phagocytic activity, mimicking primary cultured retinal microglial cells. Gene expression profile comparison between hiPSC-microglial cells and fetal/adult brain microglia revealed that hiPSC-derived microglial cells are comparable to fetal and adult microglial cells but are far away from monocytes and inflammatory monocytes (*Figure 2—figure supplements 2–4*; *Supplementary files 2 and 3*). Therefore, they effectively replicate resident microglia characteristics (*Ulland and Colonna, 2018*; *Wang et al., 2020*; *Shi et al., 2022*; *Guo et al., 2022*).

CNS disease treatment strategies include gene regulation, gene delivery (*Neumann, 2006*; *Beutner et al., 2013*), rejuvenation (*Karlstetter et al., 2015*; *Elmore et al., 2018*), or replacement of dysfunctional microglial cells (*Willis et al., 2020*; *Xu et al., 2020a*; *Han et al., 2020*; *Shibuya et al., 2022*). hiPSC-derived microglial cells offer a potentially unlimited source for cell replacement therapy. The in vivo functionality of these cells was verified through xenografting into adult Rag2−/−; IL2rg−/−;hCSF1+/+ mice. The grafted microglial cells integrated into the right location of the retina, expressing microglia signature genes and formed new homeostasis with resident microglial cells for 8 months. Homeostatic hiPSC-derived microglia responded to RPE cell injury like host retinal microglial cells, marking the success of the xenotransplantation model (*Guilliams and Scott, 2017*) and highlighting the potential for hiPSC-derived microglial cells in retinal microglia cell replacement therapy, such as in AMD.

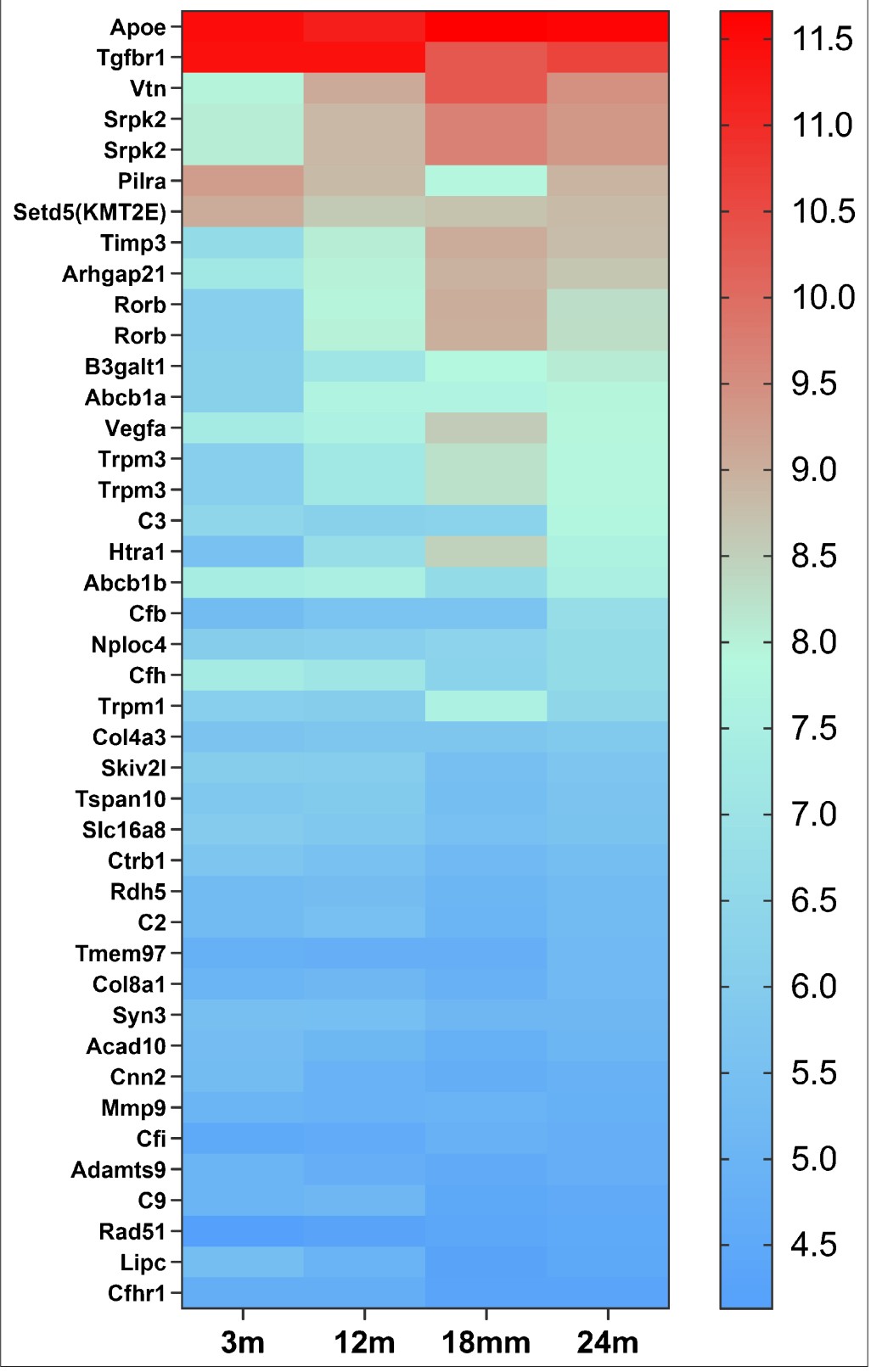

**Figure 8.** The heat map of 42 candidate genes from 34 loci associated with age-related macular degeneration (AMD) expressed in retinal microglia cells. The microglia gene expression data are from microarray data previously published (*Ma et al., 2013*). The candidate genes came from the published paper (*den Hollander et al., 2022*). The gene list is in *Supplementary file 6*.

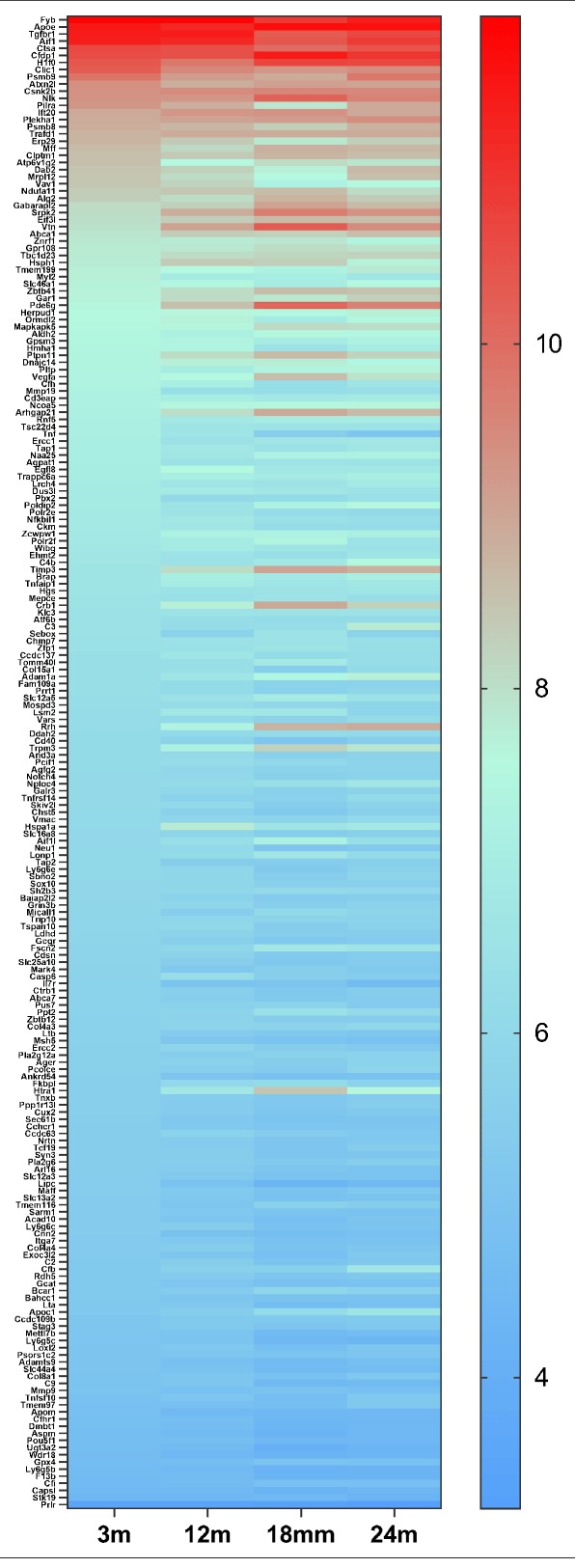

**Figure 9.** The heat map of 209 genes associated with age-related macular degeneration (AMD) (*Fritsche et al., 2016*) expressed in retinal microglia cells. The microglia gene expression data are from microarray data previously published (*Ma et al., 2013*). The gene list is in *Supplementary file 7*.

Several exogenous microglia cell replacement techniques have been investigated, with these methods varying based on the type of donor cells used and the recipient's age. Initial microglia cell replacement studies started from the transplantation of hematopoietic stem cells (HPSCs) (*Larochelle et al., 2016*; *Xu et al., 2020b*; *Hohsfield et al., 2020*). However, even after differentiation in local tissue, HPSC-derived microglial cells maintained some gene expressions distinct from the original resident microglial cells (*Lund et al., 2018*), warranting further exploration of these HPSC-derived microglial cells' unique characteristics. Another method involves using newborn mice as recipients, with iPSC or stem cell-derived microglial cells as the donors (*Mancuso et al., 2019*; *Xu et al., 2020a*). This approach does not require the depletion of resident microglial cells, and the grafted cells can infiltrate the brain tissue, distributing similarly to the original resident microglial cells. While suitable for examining microglia cell function in various backgrounds, the clinical applicability of this method remains limited.

A third transplantation approach involves adult recipients receiving iPSC or stem cell-derived microglial cells after resident microglia cell depletion (*Chadarevian et al., 2023*). Since this technique requires a microglial-empty niche, resident microglia must be depleted or relocated from the retina, typically done by using CSF1R inhibitors. To prevent these inhibitors from affecting the grafted cells, investigators have tried using modified CSF1R microglial cells as donor cells (*Chadarevian et al., 2023*). In our research, we found a 2-day recovery period with normal food intake sufficient to clear the CSF1R inhibitor, allowing grafted cells to integrate into microglia empty niche. In this study, we used the subretinal xenotransplantation method, though alternative transplantation routes such as intravenous injection, vitreous, and suborbital space delivery warrant further exploration.

In conclusion, hiPSCs can be differentiated into microglial cells through a simplified common pathway and factors, although the more precise differentiation factors still need further investigation under in vitro conditions. For instance, the microglia signature gene, TMEM119, shows low expression in hiPSC-derived microglia single-type cell culture conditions. Interestingly, we discovered that a medium composed of a mix of cultured EBs and microglia progenitor cells fosters further differentiation of microglial cells. The humanized mouse model established through xenotransplantation serves as a reliable tool for studying the functions of various hiPSC-derived microglial cells. This model offers a valuable platform for investigating disease mechanisms and evaluating the therapeutic effects of xenografted hiPSC-derived microglial cells from variety of backgrounds.

## Materials and methods

**Key resources table**

| Reagent type (species) or resource | Designation | Source or reference | Identifiers | Additional information |
|---|---|---|---|---|
| Genetic reagent (*M. musculus*) | *C;129S4-Rag2tm1.1Flv Csf1tm1(CSF1)Flv Il2rgtm1.1Flv/J* | PubMed:21791433 | MGI:J:177073 | RRID:IMSR_JAX:017708 |
| Genetic reagent (*Homo sapiens*) | *KYOUDXR0109B* | ATCC | ACS-1023 | Human-induced pluripotent stem cells (iPSCs) |
| Genetic reagent (*Homo sapiens*) | *NCRM6* | NHLBI | NCRM6 (female) iPSC line | From CD34+ cells, Episomal vectors |
| Genetic reagent (*Homo sapiens*) | *MS19-ES-H* | NHLBI | MS19-ES-H (female) iPSC line | From PBMS cells, Cytotune Sendai Virus kit |
| Genetic reagent (*Homo sapiens*) | *NCRM5-AAVS1-CAG-EGFP* | NHLBI | NCRM5-AAVS1-CAG-EGFP (clone 5) | From CD34+ cells, NCRM5 (male) reporter iPSC line with CAG-EGFP targeted mono-allelically at AAVS1 safe harbor |
| Genetic reagent (*Homo sapiens*) | *ND2-AAVS1-iCAG-tdTomato* | NHLBI | ND2-AAVS1-iCAG-tdTomato (clone 1) | From fibroblast cells, ND2 (male) reporter iPSC line with insulated CAG-tdTomato targeted mono-allelically at AAVS1 safe harbor |
| Antibody | anti-Iba1 (rabbit polyclonal) | Wako | Cat. #: 019-19741, RRID:AB_839504 | IHC (1:500) |

*Continued on next page*

*Continued*

| Reagent type (species) or resource | Designation | Source or reference | Identifiers | Additional information |
|---|---|---|---|---|
| Antibody | anti-human TMEM119 (rabbit polyclonal) | Sigma-Aldrich | Cat. #: HPA051870, RRID:AB_2681645 | IHC (1:100) |
| Antibody | anti-human CD68 (mouse monoclonal) | R&D | Cat. #: MAB20401 | IHC (1:100) |
| Antibody | anti-human CD45 (mouse monoclonal) | R&D | Cat. #: FAB1430R | IHC (1:100) |
| Antibody | anti-human CD11b (mouse monoclonal) | R&D | Cat. #: FAB1699R | IHC (1:100) |
| Antibody | anti-human CX3CR1 (rat monoclonal) | Invitrogen | Cat. #: 61-6099-42 | IHC (1:100) |
| Antibody | anti-human HLA (mouse monoclonal) | Invitrogen | Cat. #: 11-9983-42 | IHC (1:100) |
| Antibody | anti-mouse CD11b (rat monoclonal) | Bio-Rad | Cat. #: MCA711G | IHC (1:100) |
| Antibody | anti-GFAP (rat monoclonal) | Invitrogen | Cat. #: 13-0300, RRID:AB_2532994 | IHC (1:200) |
| Antibody | anti-mouse TMEM119 (guinea pig polyclonal) | Synaptic systems | Cat. #: 400 004, RRID:AB_2832239 | IHC (1:500) |
| Antibody | anti-CD68 (rat monoclonal) | Bio-Rad | Cat. #: MCA1957, RRID:AB_322219 | IHC (1:200) |
| Antibody | anti-CD34 (rat monoclonal) | eBioscience | Cat. #: 14-0341 | IHC (1:30) |
| Antibody | anti-PU.1 (rabbit monoclonal) | Thermo Fisher Scientific | Cat. #: MA5-15064 | IHC (1:200) |
| Antibody | anti-Trem2 (rabbit monoclonal) | Thermo Fisher Scientific | Cat. #: 702886 | IHC (1:100) |
| Antibody | anti-CD45 (rat monoclonal) | Bio-Rad | Cat. #: MCA1388, RRID:AB_321729 | IHC (1:100) |
| Antibody | anti glutamine synthetase (mouse monoclonal) | Millipore | Cat. #: MAB302, RRID:AB_2110656 | IHC (1:200) |
| Antibody | anti-RBPMS (guinea Pig polyclonal Ab) | Phosphosolutions | Cat. #: 1832-RBPMS | IHC (1:100) |
| Antibody | anti-cone arrestin (rabbit polyclonal) | Millipore | Cat. #: AB15282, RRID:AB_1163387 | IHC (1:200) |
| Antibody | anti-calbindin (rabbit polyclonal) | Swant | Cat. #: CB-38a | IHC (1:5000) |
| Antibody | anti-PKCa (rabbit polyclonal) | Sigma-Aldrich | Cat. #: P4334, RRID:AB_477345 | IHC (1:200) |
| Antibody | anti-RFP (rabbit polyclonal) | RockLand | Cat. #: 600-401-379-RTU | IHC (1:100) |
| Antibody | anti-Ki67-660 (rat monoclonal) | eBioscience | Cat. #: 50-5698-82, RRID:AB_2574235 | IHC (1:50) |
| Antibody | anti-P2RY12 (rabbit polyclonal) | Thermo Fisher Scientific | Cat. #: PA5-77671, RRID:AB_2736305 | IHC (1:100) |
| Antibody | anti-P2RY12 (rabbit polyclonal) | Sigma-Aldrich | Cat. #: HPA014518, RRID:AB_2669027 | IHC (1:100) |
| Antibody | Goat anti-Rabbit IgG Alexa Fluor 488 | Invitrogen | Cat. #: A27034, RRID:AB_2536097 | IHC (1:200) |
| Antibody | Goat anti-Rabbit IgG Alexa Fluor 568 | Invitrogen | Cat. #: A11011, RRID:AB_143157 | IHC (1:200) |
| Antibody | Goat anti-Rabbit IgG Alexa Fluor 647 | Invitrogen | Cat. #: A32733, RRID:AB_2633282 | IHC (1:200) |

*Continued on next page*

*Continued*

| Reagent type (species) or resource | Designation | Source or reference | Identifiers | Additional information |
|---|---|---|---|---|
| Antibody | Goat anti-mouse IgG Alexa Fluor 488 | Invitrogen | Cat. #: A28175, RRID:AB_2536161 | IHC (1:200) |
| Antibody | Goat anti-mouse IgG Alexa Fluor 568 | Invitrogen | Cat. #: A-11004, RRID:AB_141371 | IHC (1:200) |
| Antibody | Goat anti-mouse IgG Alexa Fluor 647 | Invitrogen | Cat. #: A-21235, RRID:AB_141693 | IHC (1:200) |
| Antibody | Donkey anti-Rat IgG Alexa Fluor 488 | Invitrogen | Cat. #: A-21208, RRID:AB_141709 | IHC (1:200) |
| Antibody | Donkey anti-Rat IgG Alexa Fluor 594 | Invitrogen | Cat. #: A-21209, RRID:AB_2535795 | IHC (1:200) |
| Antibody | Donkey anti-Rat IgG Alexa Fluor 650 | Invitrogen | Cat. #: SA5-10029, RRID:AB_2556609 | IHC (1:200) |
| Antibody | Rat monoclonal anti CD11b, Alexa Fluor 488 | eBioscience | Cat. #: 53-0112-82, RRID:AB_469901 | IHC (1:50) |
| Peptide, recombinant protein | human M-CSF | Invitrogen | Cat. #: PHC9501 | |
| Peptide, recombinant protein | Human IL3 | R&D | Cat. #: 203-IL-100 | |
| Peptide, recombinant protein | human IL-34 | Peprotech | Cat. #: 200–34 | |
| Peptide, recombinant protein | human CX3CL1 | Peprotech | Cat. #: 300–31 | |
| Peptide, recombinant protein | human TGFb1 | R&D | Cat. #: 7666-MB-005 | |
| Peptide, recombinant protein | human TGFb2 | R&D | Cat. #: 7346-B2-005 | |
| Peptide, recombinant protein | human BMP-4 | GIBCO | Cat. #: PHC9534 | |
| Peptide, recombinant protein | human SCF | Miltenyi Biotec | Cat. #: 130096692 | |
| Peptide, recombinant protein | human VEGF | GIBCO | Cat. #: PHC9394 | |
| Chemical compound | PLX5622 | Plexxikon | PLX5622 was provided by Plexxikon Inc and formulated in AIN-76A standard chow by Research Diets Inc | 1200 mg/kg in chow |
| Chemical compound | NaIO$_3$ | Sigma-Aldrich | Cat. #: S4007 | |
| Chemical compound | BSA | Sigma-Aldrich | Cat. #: A2153 | |
| Chemical compound | FBS | Thermo Fisher Scientific | Cat. #: A3160702 | |
| Chemical compound | Ketamine | Anased | Cat. #: NDC13985-584-10 | |
| Chemical compound | Xylazine | Anased | Cat. #: NDC59399-110-20 | |
| Chemical compound | Topical tropicamide | Alcon | Cat. # 215340 | |
| Chemical compound | Phenylephrine | Alcon | Cat. I# 215664 | |
| Chemical compound | 0.5% Proparacaine HCL | Sandoz | Cat. #: 101571 | |
| Chemical compound | Surcrose | Sigma-Aldrich | Cat. #: S7903-5KG | |
| Chemical compound | OCT | Thermo Fisher Scientific | Cat. #: 23-730-571 | |

*Continued on next page*

*Continued*

| Reagent type (species) or resource | Designation | Source or reference | Identifiers | Additional information |
|---|---|---|---|---|
| Chemical compound | Fluorescein AK-FLUOR | Akorn | Cat. #: 17478-253-10 | |
| Chemical compound | Tamoxifen | Sigma-Aldrich | Cat. #: T5648-5G | |
| Chemical compound | HBSS | Sigma-Aldrich | Cat. #: H8264-1L | |
| Chemical compound | L-(+)-Cysteine hydrochloride monohydrate | Fisher | Cat. #: C562-25 | |
| Chemical compound | Papain, lyophilized | Worthington Biochemical | Cat. #: LS003119 | |
| Chemical compound | DNAse I | Worthington Biochemical | Cat. #: LS006333 | |
| Chemical compound | Superoxide dismutase | Worthington Biochemical | Cat. #: LS003540 | |
| Chemical compound | Catalase | Sigma-Aldrich | Cat. #: C1345-1G | |
| Chemical compound | (+)-α-Tocopherol acetate | Sigma-Aldrich | Cat. #: T-1157-1G | |
| Chemical compound | Gentamicin solution | Sigma-Aldrich | Cat. #: G1397-10ml | |
| Chemical compound | D-(+)-Glucose | Sigma-Aldrich | Cat. #: G7021-100g | |
| Chemical compound | Antipain dihydrochloride | Roche | Cat. #: 11004646001 | |
| Chemical compound | HEPES | Invitrogen | Cat. #: 15630080 | |
| Chemical compound | EDTA | KD medical | Cat. #: RGC-3130 | |
| Chemical compound | RNAlater solution | Ambion | Cat. #: AM7021 | |
| Chemical compound | Triton X-100 | Sigma-Aldrich | Cat. #: X100100ml | |
| Chemical compound | Tween 20 | Sigma-Aldrich | Cat. #: P1379-100ml | |
| Chemical compound | Paraformadehyde | Fisher Scientific | Cat. #: 50-259-97 | |
| Chemical compound | Donkey serum | Sigma-Aldrich | Cat. #: D9663-10ml | |
| Chemical compound | Goat serum | Sigma-Aldrich | Cat. #: G9023-10ml | |
| Commercial assay or kit | Bloking Reagent | Sigma-Aldrich | Cat. #: 11096176001 | |
| Commercial assay or kit | In Situ Cell Death Detection Kit, TMR red | Sigma-Aldrich | Cat. #: 12156792910 | |
| Commercial assay or kit | Ib4 Alexa Fluor 568 | Invitrogen | Cat. #: I21412 | IHC (1:200) |
| Commercial assay or kit | Ib4 Alexa Fluor 647 | Invitrogen | Cat. #: I32450 | IHC (1:200) |
| Commercial assay or kit | DAPI | Sigma-Aldrich | Cat. #: D9542 | IHC (1:200) |
| Commercial assay or kit | Mounting medium without DAPI | Vector | Cat. #: H-1000 | |
| Commercial assay or kit | Mounting medium with DAPI | Vector | Cat. #: H-1200 | |
| Commercial assay or kit | RNeasy Mini Kit | QIAGEN | Cat. #: 74104 | |
| Commercial assay or kit | Rnase free Dnase set | QIAGEN | Cat. #: 79254 | |
| Commercial assay or kit | First strand cDNA synthesis | Takara | Cat. #: 6110A | |
| Commercial assay or kit | MessageBooster cDNA synthesis kit | Epicentre | Cat. #: MB060110 | |
| Commercial assay or kit | Fast SYBR Green Master Mix | Thermo Fisher Scientific | Cat. #: 4385617 | |

*Continued on next page*

*Continued*

| Reagent type (species) or resource | Designation | Source or reference | Identifiers | Additional information |
|---|---|---|---|---|
| Commercial assay or kit | LiDirect Lightening genotyping kit | LifeSci | Cat. #: M0015 | |
| Commercial assay or kit | eBioscience Flow Cytometry Staining Buffer | Thermo Fisher Scientific | Cat. #: 00-4222-57 | |
| Commercial assay or kit | X-VIVO-15 medium | Lonza | Cat. #: BEBP02-061Q | |
| Commercial assay or kit | DMEM:F12 medium | Thermo Fisher Scientific | Cat. #: 11330057 | |
| Commercial assay or kit | mTeSR1 | Stemcell technologies | Cat. #: 85850 | |
| Commercial assay or kit | N2 supplement | Thermo Fisher Scientific | Cat. #: 17502048 | |
| Commercial assay or kit | Non-essential Amino Acids (NEAA) | Thermo Fisher Scientific | Cat. #: 11140050 | |
| Commercial assay or kit | GlutaMax Supplement | Thermo Fisher Scientific | Cat. #: 35050061 | |
| Commercial assay or kit | Geltrex | Thermo Fisher Scientific | Cat. #: A1413301 | |
| Commercial assay or kit | TrypLE Express | Thermo Fisher Scientific | Cat. #: 12605010 | |
| Commercial assay or kit | Rho-kinase inhibitor Y-27632 | abcam | Cat. #: ab143784 | |
| Commercial assay or kit | mFreSR | Stemcell technologies | Cat. # 05854 | |
| Commercial assay or kit | Stem Cell Dissociation Reagent | ATCC | Cat, #: ACS-3010 | |
| Commercial assay or kit | Stem Cell Freezing Media | ATCC | Cat. #: ACS-3020 | |
| Commercial assay or kit | Penicillin-Streptomycin | Thermo Fisher Scientific | Cat. #: 15140122 | |
| Commercial assay or kit | pHrodo Red *E. coli* BioParticles | Thermo Fisher Scientific | Cat. #: P35361 | |
| Commercial assay or kit | pHrodo Red Zymosan Bioparticles | Thermo Fisher Scientific | Cat. #: P35364 | |
| Commercial assay or kit | Bovine rod outer segment | Invision Bioresources | Cat. #: 98740 | |
| Commercial assay or kit | Vybrant DiI Cell-Labeling Solution | Thermo Fisher Scientific | Cat. #: V22885 | |
| Commercial assay or kit | Lipopolysaccharides (LPS) | Sigma-Aldrich | Cat. #: L2630 | |
| Commercial assay or kit | RIPA lysis buffer | Sigma-Aldrich | Cat. #: R0278 | |
| Commercial assay or kit | proteinase inhibitor mixture | Calbiochem | Cat. #: 539132 | |
| Commercial assay or kit | Pierce BCA Protein Assay Kit | Thermo Fisher Scientific | Cat. #: 23227 | |
| Commercial assay or kit | Milliplex bead assay kit | Millipore | Cat. #: MCYTOMAG-70K | |
| Commercial assay or kit | AggreWellsTM800 | Stemcell technologies | Cat. #: 34825 | |
| Software | ImageJ | ImageJ (http://imagej.nih.gov/ij/) | RRID:SCR_003070 | |
| Software | GraphPad Prism7 | GraphPad Prism (https://graphpad.com) | RRID:SCR_015807 | Version 7 |
| Software | IPA | QIAGEN | RRID:SCR_008653 | |
| Software | JMP | JMP | RRID:SCR_014242 | Version 12 |

## Experimental animals and PLX-5622 treatment

In vivo experiments were conducted according to protocols (NEI-698) approved by the Institutional Animal Care and Use Committee (National Eye Institute Animal Care and Use Committee) and adhered to the Association for Research in Vision and Ophthalmology (ARVO) statement on animal use in ophthalmic and vision research. Rag2$^{-/-}$;IL2rg$^{-/-}$;hCSF1$^{+/+}$ transgenic mice were obtained from

Jackson Laboratories (Stock #17708). Animals were housed in a National Institutes of Health (NIH) animal facility under a 12-hr light/dark cycle and fed standard chow. Genotype analysis by sequencing revealed that the rd8 mutation was absent in the *Crb1* gene. To deplete retinal microglia, 2-month-old experimental animals were administered a diet containing PLX-5622 (Plexxikon, at 1200 parts per million), a potent and selective inhibitor of the CSF1R previously demonstrated to deplete the majority of microglia in the mouse brain (*Dagher et al., 2015*) and retina (*Zhang et al., 2018*). Animals were maintained continuously on the PLX-5622 diet for 10 days and then switched back to standard chow.

## Human iPSC culture

Five human iPSC lines were used in this study. The KYOUDXR0109B hiPSC line was generated in Yamanaka Lab from fibroblasts isolated from a healthy female donor and reprogrammed by the expression of OCT4, SOX2, KLF4, and MYC using retroviral transduction. Cells are tested for post-freeze viability and growth, sterility (including mycoplasma), identity by short-tandem repeat (STR) analysis and karyotype by G-banding. Each lot is tested for pluripotency using flow cytometry for the expression of the pluripotent markers (KYOUDXR0109B Human Induced Pluripotent Stem (IPS) Cells [201B7] (ATCC ACS1023)). The line of NCRM5-AAVS1-CAG-EGFP (clone 5), ND2-AAVS1-iCAG-tdTomato (clone 1), NCRM6, and MS19-ES-H were obtained from the NHLBI iPSC Core Facility of National Heart, Lung and Blood Institute (NHLBI). NCRM5-AAVS1-CAG-EGFP is an EGFP-expressing reporter iPSC line with CAG-EGFP targeted mono-allelically at the AAVS1 safe harbor locus in NCRM5 iPSCs. These iPSCs were reprogrammed from CD34$^+$ peripheral blood mononuclear cells (PBMCs) from a healthy male donor and the cell line was authenticated (*Luo et al., 2014*). ND2-AAVS1-iCAG-tdTomato is a tdTomato-expressing reporter iPSC line with insulated CAG-tdTomato targeted mono-allelically at the AAVS1 safe harbor locus in ND2 iPSCs. These ND2-AAVS1-iCAG-tdTomato (clone1) cell line was reprogrammed using healthy male fibroblast cells and was authenticated by STR profiling performed by WiCell Cytogenetics lab using a Powerplex 16 System (Promega) and genomic DNA extracted from the iPSCs with DNeasy Blood and Tissue Kit (QIAGEN) (*Patterson et al., 2020*). Microglia differentiated from these two reporter lines were used for the xenotransplantation experiments. The MS19-ES-H line was reprogrammed from PBMCs from a healthy female donor with a Cytotune Sendai Virus kit (Thermo Fisher). The STR profiling was performed by WiCell Cytogenetics lab using a Powerplex 16 System (Promega) and genomic DNA extracted from the iPSCs with DNeasy Blood and Tissue Kit (QIAGEN) (*Patterson et al., 2019*). NCRM6 iPSC line was reprogrammed from CD34$^+$ PBMCs from a healthy female donor with episomal iPSC reprogramming vectors (Thermo Fisher). This cell line was authenticated by Cell Line Genetics for STR service using the above same method. All hiPSC lines used in this study were determined by mycoplasma with MycoAlert (Lonza's MycoAlert Plus kit) and excluded mycoplasma contamination.

Cells were cultured on Geltrex-coated (0.2 mg/ml, Gibco, #A1413302) 6-well plates using 1× mTeSRTM-1 medium. Passaging was performed following dissociation with TrypLE Express enzymatic digestion (Gibco by Life Technologies). Upon initial plating, cells were cultured in medium containing 3 µM Rho-kinase inhibitor (Y-27632, Abcam). The medium was completely refreshed every day. Cells reaching 70% confluence were either passaged or cryopreserved in Stem Cell Freezing Media (mFreSR, StemCells, Catalog # 05854).

## Myeloid progenitor cell differentiation and microglial cell maturation

The protocol for the differentiation of hiPSC cells to myeloid progenitors and then to microglia were adapted and modified from those described previously by the Cowley laboratory (*van Wilgenburg et al., 2013*; *Haenseler et al., 2017*). The first step of the protocol to enable EB formation employed the Spin-EBs formation method performed in AggreWellsTM800 microwell culture plates (Stemcell Technologies, Catalog # 34825). Briefly, 1 ml of mTeSRTM-1 spin-EB medium was added into each culture well and centrifuged at 3000 × *g* for 2 min. Subsequently, 4 × 10$^6$ iPSCs in 1 ml of medium were added to the well of spin-EB plate and then centrifuged at 800 rpm for 3 min. The plate was incubated at 37°C, 5% CO$_2$ for 4 days, then 1 ml medium was replaced in a drop-wise manner every day for the next 4 days with an EB medium: mTeSR1 medium (STEMCELL Technologies) containing 50 ng/ml BMP-4 (Gibco- PHC9534), 20 ng/ml human stem cell factor (SCF, Miltenyi Biotec), and 50 ng/ml vascular endothelial growth factor (hVEGF, Gibco- PHC9394).

In the second step of myeloid progenitor differentiation, the resulting EBs were harvested with a 40-µm filter column. Approximately 150–200 EBs were transferred into a 75-cm² flask containing myeloid cell differentiation medium: TheraPEAK X-VIVO-15 Serum-free Hematopoietic Cell Medium (Lonza, Cat#: BEBP04-744Q) containing 100 ng/ml M-CSF (Invitrogen), 25 ng/ml IL-3 (R&D), 2 mM Glutamax supplement (Invitrogen), 1× N2 supplement (Thermo Fisher Scientific, Cat#17502048). Two-thirds of the media volume in the culture flask was replaced every 5 days for 2–3 weeks.

In the third step of microglia differentiation, the non-adherent floating cell layer consisting of differentiated myeloid progenitor cells was harvested from the supernatant and transferred into the 6-well plate and allowed to settle and adhere overnight. Two-thirds of the medium volume was removed and replaced with a microglia cell differentiation medium: DMEM/F12 medium (Gibco, #11330) containing 50 ng/ml M-CSF (Invitrogen), 100 ng/ml IL-34 (Peprotech), 10 ng/ml TGFb1 and 2–5 ng/ml TGFb2 (both from R&D Systems), 20 ng/ml CX3CL1 (Peprotech), 1× N2 supplement and 2 mM Glutamax supplement. The culture was maintained for 2 weeks after which the differentiated iPSC-derived microglia were harvested for analysis or for cell transplantation.

## Phagocytosis assay

To assess microglia phagocytosis in vitro, the following bioparticles were employed as targets: (1) pHrodo Red *E. coli* BioParticles Conjugate for Phagocytosis (Thermo Fisher Scientific, Cat#P35361); (2) pHrodo Red Zymosan Bioparticles Conjugate for Phagocytosis (Thermo Fisher Scientific, Cat#P35364); (3) bovine rod POSs (Invision Bioresources, Cat#98740). Bovine POS was diluted in serum-free DMEM/F12 (1:1; Gibco) to a concentration of $10^6$ segments/ml and fluorescently labeled with the lipophilic dye DiI (Vybrant Cell-Labeling Solutions; Invitrogen) according to the manufacturer's instructions. Each phagocytosis assay employed $1 \times 10^5$ POSs and 2 mg/ml pHrodo Red *E. coli* membrane/Zymosan BioParticles.

For the assay, harvested floating myeloid cells were transferred into 4-well chamber slides (Thermo Fisher). For phagocytosis assessment of myeloid cells, the cells were cultured for 1 day and challenged with bioparticles. To assess iPSC-derived microglia, the cells were cultured in a microglia cell differentiation medium for 2 weeks and then challenged. In the assay, bioparticles were added to the 100 µl serum-free DMEM/F12 medium in the slide chamber, incubated for 1 hr at 37°C, washed three times with phosphate-buffered saline (PBS), and then fixed in 4% paraformaldehyde (PFA) for 20 min. Fixed cells were immunostained with antibodies to IBA1 and P2RY12 and counterstained with DAPI. Stained cells were imaged with an Olympus 1000 confocal microscope, and image analysis was conducted using ImageJ software (NIH).

## mRNA and protein analysis following LPS challenge in vitro

Differentiated hiPSC-derived microglia cultured in 6-well plates were stimulated with LPS at 100 ng/ml for 6 and 24 hr. Microglia stimulated for 6 hr were collected in RNAlater solution (Thermo Fisher) and stored at −80°C for further qRT-PCR analysis. Microglia stimulated for 24 hr were collected for protein quantification. After the medium was collected, the cells in the well were washed with 1× PBS, and then 200 µl of RIPA lysis buffer with proteinase inhibitor cocktail (Calbiochem) was added; the cells were removed by scraping, collected into 1.5 ml Eppendorf tube, and then homogenized with sonication (Sonicator 125 Watts, Qsonica) at 4°C. After sonication and centrifugation, total protein concentration was measured (BCA protein assay kit; Pierce). Levels of individual cytokines were determined using a Milliplex bead assay kit (Milliplex MAP human cytokine/chemokine magnetic bead panel, #MCYTOMAG-70K; Millipore) and involving the Luminex MAPIX system with data analysis using xPONENT 4.2 software (Luminex). The cytokines analyzed included *IL1A*, *IL1B*, *IL6*, *IL8*, *TNFa*, *CXCL10*, *CCL2*, *CCL3*, *CCL4*, and *IL10*.

## mRNA expression analysis by quantitative RT-PCR

mRNA expression was quantitated using qRT-PCR. Harvested cells were lysed by trituration and homogenized using QIAshredder spin columns (QIAGEN). Total RNA was isolated using the RNeasy Mini kit (QIAGEN) according to the manufacturer's specifications. First-strand cDNA synthesis from mRNA was performed using qScript cDNA SuperMix (Quanta Biosciences) using oligo-dT as primer. qRT-PCR was performed using an SYBR green RT-PCR kit (Affymetrix), using the Bio-Rad CFX96 Touch Real-Time PCR Detection System under the following conditions: denaturation at 95°C for 5 min,

followed by 40 cycles of 95°C for 10 s, and then 60°C for 45 s. Threshold cycle (CT) values were calculated and expressed as fold-induction determined using the comparative CT (2ΔΔCT) method. Ribosomal protein S13 (RPS13) and GAPDH were used as internal controls. Oligonucleotide primers are provided in *Supplementary file 4*.

## Transplantation of hiPSC-derived microglia by subretinal injection

Differentiated hiPSC-derived microglia grown in flasks were first washed in 1× PBS before being removed by scraping and collected into a 50-ml tube in 5 ml PBS. Cell numbers were counted using a cell counter (Countess 3, Thermo Fisher). Microglia were collected by centrifugation (5 min at 4°C, 200 × $g$) and the resulting cell pellet resuspended in 1× PBS at a concentration of 5000 cells/μl for in vivo transplantation via subretinal injection. Experimental animals were given general anesthesia (ketamine 90 mg/kg and xylazine 8 mg/kg) and additional topical anesthesia (0.5% Proparacaine HCL, Sandoz) applied to the injected eye. For the injection, the temporal sclera was exposed by a conjunctival cut-down and a scleral incision made 0.5 mm behind the limbus using 32 G needle to access the subretinal space. The tip of a blunt 32 G needle attached to a Hamilton micro-syringe was introduced through the incision at an angle 5 degrees tangent to the globe and advanced 0.5–1 mm into the subretinal space under a dissecting microscope. Microglial cells (5000 cells in 1 μl PBS) were slowly injected from the micro-syringe into the subretinal space using an aseptic technique. Post-procedure, treated eyes were carefully inspected for signs of bleeding or distention and intraocular pressure was monitored using a tonometer (iCare TONOLAB, Finland). In the event that intraocular pressure remained elevated (>20 mmHg) and/or the globe appeared distended, a vitreous tap was performed using a 33 G needle to reduce intraocular pressure. In the unlikely event that excessive bleeding is observed, the animal will be examined by a veterinarian or euthanized immediately.

## Immunohistochemical analyses

For immunohistochemical analysis of microglia in vitro, microglia were differentiated in 4-well chambered slides, fixed in 4% PFA for 20 min and processed for immunostaining. For in vivo analyses, recipient animals were euthanized by $CO_2$ inhalation, and the eyes were enucleated. Enucleated eyes were dissected to form posterior segment eyecups and fixed in 4% PFA in phosphate buffer (PB) for 2 hr at 4°C. Eyecups were either cryosectioned (Leica CM3050S) or further dissected to form retinal flat mounts. Flat-mounted retinas were blocked for 1 hr in a blocking buffer containing 10% normal donkey serum and 1% Triton X-100 in PBS at room temperature. Primary antibodies included IBA1 (1:500, Wako, #019-19741), anti-mouse Tmem119 (1:500, Synaptic Systems, #400 004), anti-human TMEM119 (1:100, Sigma, #HPA051870), anti-mouse Cd68 (1:200, Bio-Rad, #MCA1957), anti-human CD68 (1:100, R&D, #MAB20401), anti-mouse Cd45 (1:100, Bio-Rad, #MCA1388), anti-human CD45 (1:100, R&D, #FAB1430R), cone arrestin (1:200, Millipore, #AB15282), Ki67 (1:30, eBioscience, #50-5698-82), anti-P2RY12 (1:100, Thermo Fisher, #PA5-77671 and Sigma, #HPA014518), CD34 (1:50, eBioscience, #14–0341), hCD11b (1:100, R&D, #FAB1699R), mCD11b (Bio-Rad, Cat#: MCA711G, 1:100), CX3CR1 (1:100, Invitrogen, #61-6099-42), hHLA (Invitrogen, #11-9983-42), SPI1 (Invitrogen, #MA5-15064), TREM2 (1:100, Invitrogen, #702886), glutamine synthetase (1:200, Millipore, #MAB302), PKCa (1:200, Sigma-Aldrich, #p4334), GFAP (1:200, Invitrogen, #13-0300), RBPMS (1:100, Phosphosolutions, 1832-RBPMS), Calbindin (1:5000, Swant, CB-38a), and anti-RFP (1:200, RockLand, 600-401-379-RTU). Primary antibodies were diluted in blocking buffer and incubated at 4°C overnight for retinal sections and at room temperature overnight on a shaker for retinal flat mounts. Experiments in which primary antibodies were omitted served as negative controls. After washing in 1× PBST (0.2% Tween-20 in PBS), retinal samples were incubated for 2 hr at room temperature with secondary antibodies (AlexaFluor 488-, 568-, or 647-conjugated anti-rabbit, mouse, rat, goat, and guinea pig IgG) and DAPI (1:500; Sigma-Aldrich) to label cell nuclei. Isolectin B4 (IB4), conjugated to AlexaFluor 568/647 (1:100, Life Technologies), was used to label activated microglia and retinal vessels. Stained retinal samples were imaged with confocal microscopy (Olympus FluoView 1000, or Zeiss LSM 880, or Nikon A1R). For analysis at high magnification, multiplane z-series were collected using 20 or 40 objectives; Confocal image stacks were viewed and analyzed with FV100 Viewer Software, NIS-Element Analysis and ImageJ (NIH).

## RNAseq analysis

For whole transcriptome analysis, cultured myeloid progenitor cells and differentiated microglial cells with and without 0.1 µg/ml LPS treatment were harvested in the flasks and 6-well plates, respectively. After harvesting, all samples stored frozen in RNAlater (Roche) solution before RNA extraction using the QIAGEN RNA Mini Kit. RNA quality and quantity were evaluated using Bioanalyzer 2100 with the RNA 6000 Nano Kit (Agilent Technologies). The preparation of RNA library and transcriptome sequencing was performed using an external vendor (Novogene, Sacramento, CA). Genes with adjusted p-value <0.05 and log2FC (Fold Change) >1 were considered differentially expressed. IPA (QIAGEN) was employed for canonical pathway and graphical pathway analysis. The microglia gene list was constructed from our previous microarray data from retinal microglial cells and published data (*Ma et al., 2013*; *Bennett et al., 2016*). The heat map, volcano, and histogram plot were performed using Prism 9.5.1 (GraphPad).

## NaIO$_3$-induced model of RPE cell injury

Recipient mice 8 months following xenotransplantation were administered a single dose of NaIO$_3$ (Honeywell Research Chemicals) at a dose of 30 mg/kg body weight via intraperitoneal injection. Animals were euthanized 3 and 7 days after NaIO$_3$ injection, and their retinas were harvested and subjected to histological and molecular analysis.

## Statistics and reproducibility

All data in the graphical panel represent mean ± standard error. When only two independent groups were compared, significance was determined by a two-tailed unpaired *t*-test with Welch's correction. When three or more groups were compared, one-way ANOVA with the Bonferroni post hoc test or two-way ANOVA was used. A p-value <0.05 was considered significant. The analyses were done in GraphPad Prism v.5. All experiments were independently performed at least three experimental replicates to confirm consistency in observations across replicates.

## Acknowledgements

This study was supported by the National Eye Institute Intramural Research program.

## Additional information

### Funding

| Funder | Grant reference number | Author |
|---|---|---|
| National Eye Institute | | Wenxin Ma<br>Lian Zhao<br>Biying Xu<br>Robert N Fariss<br>T Michael Redmond<br>Jizhong Zou<br>Wai T Wong<br>Wei Li |

The funders had no role in study design, data collection, and interpretation, or the decision to submit the work for publication.

### Author contributions

Wenxin Ma, Conceptualization, Data curation, Formal analysis, Validation, Methodology, Writing – original draft, Writing – review and editing; Lian Zhao, Data curation, Methodology; Biying Xu, Data curation; Robert N Fariss, Writing – review and editing; T Michael Redmond, Supervision; Jizhong Zou, Resources, Writing – review and editing; Wai T Wong, Conceptualization, Supervision, Writing – review and editing; Wei Li, Conceptualization, Supervision, Visualization, Writing – review and editing

### Author ORCIDs

Wenxin Ma (iD) https://orcid.org/0000-0001-8396-6625

Biying Xu https://orcid.org/0000-0001-8608-6797
Wai T Wong http://orcid.org/0000-0003-0681-4016
Wei Li https://orcid.org/0000-0002-2897-649X

## Ethics

This study was performed in strict accordance with the recommendations in the Guide for the Care and Use of Laboratory Animals of the National Institutes of Health. All of the animals were handled according to approved Institutional Animal Care and Use Committee (IACUC) protocols (#NEI-698) of approved by the Institutional Animal Care and Use Committee (National Eye Institute Animal Care and Use Committee) and adhered to the Association for Research in Vision and Ophthalmology (ARVO) statement on animal use in ophthalmic and vision research. All surgery was performed under sodium pentobarbital anesthesia, and every effort was made to minimize suffering.

Reviewer #1 (Public Review): https://doi.org/10.7554/eLife.90695.3.sa1
Reviewer #2 (Public Review): https://doi.org/10.7554/eLife.90695.3.sa2
Author response https://doi.org/10.7554/eLife.90695.3.sa3

## Additional files

### Supplementary files

• Supplementary file 1. Microglia-enriched gene list. The gene list of 71 microglia-enriched genes was extracted from bulk RNAseq of microglial cells vs. myeloid progenitor cells (MPCs). The total microglia-enriched gene list combined from the research conducted by Barres BA Lab (Bennett et al., Proc Natl Acad Sci U S A, 2016) and from our RNA sequencing data of mouse retinal microglia (*Ma et al., 2013*), identifying a total of 130 genes predominantly expressed in microglia.

• Supplementary file 2. 188 hMG gene comparisons vs. GSE178846. The human microglia gene panel combined our mouse microglia-enriched genes and human microglia-enriched genes (*Abud et al., 2017*; *Muffat et al., 2016*; *Douvaras et al., 2017*; *Böttcher et al., 2019*; *van der Poel et al., 2019*). The total microglia-enriched gene list contains 203 genes, which used to be extracted from the gene profile of hiPSC-MG and downloaded human adult microglia (AMG), fetal microglia (FMG), inflammatory monocyte (IM) and monocytes (M) (GSE 178846, *Abud et al., 2017*). 188 genes were obtained from both gene lists. All the gene counts were normalized with four human cells housekeeping genes C1orf43, RAB7A, REEP5, and VCP (*Eisenberg and Levanon, 2013*).

• Supplementary file 3. The gene list of comparison vs. GSE111972. The total gene list of both male and female hiPSC-derived microglial cells and human brain microglia cells (GSE 111972, *van der Poel et al., 2019*).

• Supplementary file 4. The oligos for quantitative reverse transcription-PCR (qRT-PCR). The oligos were used for qRT-PCR in cultured hiPSC-derived cells.

• Supplementary file 5. Human-specific oligos for quantitative reverse transcription-PCR (qRT-PCR). The oligos were used for qRT-PCR in mouse retinas with integrated hiPSC-derived microglial cells.

• Supplementary file 6. 42 genes associated with age-related macular degeneration (AMD). 42 candidate genes from 34 loci associated with AMD expressed in retinal microglia cells. The microglia gene expression data are from microarray data previously published (*Ma et al., 2013*). The candidate genes came from the published paper (*den Hollander et al., 2022*).

• Supplementary file 7. 209 genes associated with age-related macular degeneration (AMD). The 209 microglia-enriched genes were extracted from the genes associated with AMD (*Fritsche et al., 2016*). The microglia gene expression data are from microarray data previously published (*Ma et al., 2013*).

• MDAR checklist

### Data availability

All data generated or analyzed during this study are included in the manuscript and supporting files; source data files have been provided for Figures 2 and 3 and Figure 2 suppl. A, B, and C.

The following previously published datasets were used:

| Author(s) | Year | Dataset title | Dataset URL | Database and Identifier |
|---|---|---|---|---|
| Abud EM, Ramirez RN, Martinez ES, Healy LM, Nguyen CHH, Newman SA, Yeromin AV, Scarfone VM, Marsh SE, Fimbres C, Caraway CA, Fote GM, Madany AM, Agrawal A, Kayed R, Gylys KH, Cahalan MD, Cummings BJ, Antel JP, Mortazavi A, Carson MJ, Poon WW, Blurton-Jones M | 2017 | Generation of human microglia-like cells to study neurological disease | https://www.ncbi.nlm.nih.gov/geo/query/acc.cgi?acc=GSE89189 | NCBI Gene Expression Omnibus, GSE89189 |
| an der Poel M, Ulas T, Mizee MR, Hsiao CC, Miedema SSM, Adelia SKG, Helder B, Tas SW, Schultze JL, Hamann J, Huitinga I | 2019 | Transcriptional profiling of human microglia reveals grey-white matter heterogeneity and multiple sclerosis-associated changes | https://www.ncbi.nlm.nih.gov/geo/query/acc.cgi?acc=GSE111972 | NCBI Gene Expression Omnibus, GSE111972 |

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
