## [Editor Report · eLife assessment]

The authors have improved a method to differentiate human iPSC-derived microglial cells with immune responses and phagocytic abilities; and through transplantation into the adult mouse retina, the authors further demonstrated their integration and occupation of native microglial cell space, and functional response to retinal injuries. The study is **important** and the data are **convincing** for potential microglial replacement therapy to treat retinal and CNS diseases.

---

## [Referee Report · Reviewer #1 (Public Review)]

Summary:

This paper reported a protocol of using human-induced pluripotent stem cells to generate cells expressing microglia-enriched genes and responding to LPS by drastically upregulation of proinflammatory cytokines. Upon subretinal transplantation in mice, hiPSC-derived cells integrated into the host retina and maintained retinal homeostasis while they responded to RPE injury by migration, proliferation, and phagocytosis. The findings revealed the potential of using hiPSC-derived cell transplantation for microglia replacement as a therapeutic strategy for retinal diseases.

Strengths:

The paper demonstrates a method of consistently generating a significant quantity of hiPSC-derived microglia-like cells for in vitro study or for in vivo transplantation. RNAseq analysis offers an opportunity for comprehensive transcriptome profiling of the derived cells. It is impressive that following transplantation, these cells were well integrated into the retina, migrated to the corresponding layers, adopted microglia-like morphologies, and survived for a long term without generating apparent harm. The work has laid a foundation for future utilization of hiPSC-derived microglia in lab and clinical applications.

Weaknesses:

(1) The primary weakness of the paper concerns its conclusion of having generated "homogenous mature microglia", partly based on the RNAseq analysis. However, the comparison of gene profiles was carried out only between "hiPSC-derived mature microglia" and the proliferating myeloid progenitors. While the transcriptome profiles revealed a trend of enrichment of microglia-like gene expression in "hiPSC-derived mature microglia" compared to proliferating myeloid progenitors, this is not sufficient to claim they are "mature microglia". It is important that one carries out a comparative analysis of the RNAseq data with those of primary human microglia, which may be done by leveraging the public database. To convincingly claim these cells are mature microglia, questions to be addressed include how similar the molecular signatures of these cells are compared with the fully differentiated primary microglia cell or if they remain progenitor-like or take on mosaic properties, and how they distinguish from macrophages.

(2) While the authors attempted to demonstrate the functional property of "hiPSC-derived mature microglia" in culture, they used LPS challenge, which is an inappropriate assay. This is because human microglia respond poorly to LPS alone but need to be activated by a combination of LPS with other factors, such as IFNγ. Their data that "hiPSC-derived mature microglia" showed robust responses to LPS indeed implicates that these cells do not behave like mature human microglia.

(3) The resolution of Figs. 4 - 6 is so low that even some of the text and labels are hardly readable. Based on the morphology shown in Fig. 4 and the statement in line 147, these hiPSC-derived "cells altered their morphology to a rounded shape within an hour of incubation and rapidly internalized the fluorescent-labeled particles". This is a peculiar response. Usually, microglia do not respond to fluorescent-labeled zymosan by turning into a rounded-shaped morphology within an hour when they internalize them. Such a behavior usually implicates weak phagocytotic capacity.

(4) Data presented in Fig. 5 are not very convincing to support that transplanted cells were immunopositive for "human CD11b (Fig.5C), as well as microglia signature markers P2ry12 and TMEM119 (Fig.5D)" (line 167). The resolution and magnification of Fig. 5D are too low to tell the colocalization of tdT and human microglial marker immunolabeling. In the flat-mount images (C, I), hCD11b immunolabeling is not visible in the GCL or barely visible in the IPL. This should be discussed.

(5) Microglia respond to injury by becoming active and losing their expression of the resting state microglial marker, such as P2ry12, which is used in Fig. 6 for the detection of migrated microglia. To confirm that these cells indeed respond to injury like native microglia, one should check for activated microglial markers and induction of pro-inflammatory cytokines in the sodium iodate-injury model.

---

## [Referee Report · Reviewer #2 (Public Review)]

Summary:

Ma et al. employed a myeloid progenitor/microglia differentiation protocol to produce human-induced pluripotent stem cell (hiPSC)-derived microglia in order to examine the potential of microglial cell replacement as a treatment for retinal disorders. They characterized the iPSC-derived microglia by gene expression and in vitro assay analysis. By evaluating xenografted microglia in the partly microglia-depleted retina, the function of the microglia was further assessed.

Overall, the study and the data are convincing, and xenografted microglia were also tested in a RPE injury paradigm.

---

## [Author Response]

The following is the authors’ response to the original reviews.

**Reviewer #1 (Public review):**
(1) The primary weakness of the paper concerns its conclusion of having generated "homogenous mature microglia", partly based on the RNAseq analysis. However, the comparison of gene profiles was carried out only between "hiPSC-derived mature microglia" and the proliferating myeloid progenitors. While the transcriptome profiles revealed a trend of enrichment of microglia-like gene expression in "hiPSC-derived mature microglia" compared to proliferating myeloid progenitors, this is not sufficient to claim they are "mature microglia". It is important that one carries out a comparative analysis of the RNAseq data with those of primary human microglia, which may be done by leveraging the public database. To convincingly claim these cells are mature microglia, questions need to be addressed including how similar the molecular signatures of these cells are compared with the fully differentiated primary microglia cell or if they remain progenitor-like or take on mosaic properties, and how they distinguish from macrophages.

We greatly appreciate the insightful comments and suggestions from the reviewers, which were instrumental in enhancing our data analysis and organization. In response to the feedback, we have updated the terminology from “mature microglia” to simply “microglia” while clarifying in our text that these are fully differentiated microglia under single-type cell culture conditions.

Guided by the reviewer's advice, we incorporated RNA-seq data from human brain microglia studies conducted by Dr. Poon and Dr. Blurton-Jones' Lab (Abud et al., Neuron, 2017) and Dr. Huitinga's Lab (van der Poel et al., Nat Commun, 2019). We then conducted a comparative analysis of the gene expression profiles between our fully differentiated hiPSC-derived microglia and those from fetal/adult brain microglia (see Fig.2. Suppl. B, C and D; Suppl. table 1 and table 2). The correlation analysis revealed that our hiPSC-derived microglia closely resemble fetal and adult brain microglia, distinguishing them significantly from monocytes and inflammatory monocytes.

(2) While the authors attempted to demonstrate the functional property of "hiPSC-derived mature microglia" in culture, they used LPS challenge, which is an inappropriate assay. This is because human microglia respond poorly to LPS alone but need to be activated by a combination of LPS with other factors, such as IFNγ. Their data that "hiPSC-derived mature microglia" showed robust responses to LPS indeed implicates that these cells do not behave like mature human microglia.

We appreciate the feedback received. In response, we cultured hiPSC-derived microglia cells and subjected them to treatments with IFNγ, LPS, and a combination of both IFNγ+LPS, as illustrated in Figure 3 suppl. Our findings revealed that the IFNγ+LPS combination notably enhanced the expression of IL1a, IL1b, TNFa, CCL8, and CXCL10, whereas IL6 and CCL2 levels remained unchanged. Treatment with IFNγ alone significantly elevated the expression of TNFa, CCL8, CXCL10, and CCL2. These outcomes align with the findings reported by Rustenhoven et al. (Sci Rep, 2016), suggesting that the functionality of our hiPSC-derived microglia cells closely mirrors that of primary human adult microglia cells.

(3) The resolution of Figs. 4 - 6 is so low that even some of the text and labels are hardly readable. Based on the morphology shown in Fig. 4 and the statement in line 147, these hiPSC-derived "cells altered their morphology to a rounded shape within an hour of incubation and rapidly internalized the fluorescent-labeled particles". This is a peculiar response. Usually, microglia do not respond to fluorescent-labeled zymosan by turning into a rounded shaped within an hour when they internalize them. Such a behavior usually implicates weak phagocytotic capacity.

Thank you for your insightful comments. During submission, the main text's PDF version was converted online, resulting in low-quality output. We have since updated this with a high-resolution version. The observed alterations in cell morphology following zymosan phagocytosis may be attributed to the high zymosan concentration used (2mg/ml). We conducted an assessment to understand the impact of zymosan concentration on the morphology of hiPSC-derived microglial cells, as shown in Figure 4 suppl B. Our findings indicate that microglia cells adopt an amoeboid, rounded shape at zymosan concentrations exceeding 20ug/ml. To clarify this point, we have amended the text to read: "The cells altered their morphology and rapidly internalized the fluorescent-labeled particles."

(4) Data presented in Fig. 5 are not very convincing to support that transplanted cells were immunopositive for "human CD11b (Fig.5C), as well as microglia signature markers P2ry12 and TMEM119 (Fig.5D)" (line 167). The resolution and magnification of Fig. 5D is too low to tell the colocalization of tdT and human microglial marker immunolabeling. In the flat-mount images (C, I), hCD11b immunolabeling is not visible in the GCL or barely visible in the IPL. This should be discussed.

We are grateful for the reviewer's comments. As previously mentioned, the low quality of the images was due to the online conversion of the PDF version. We have now submitted both high-quality PDF and Word versions for the reviewer's assessment. In these high-quality versions, the colocalization of tdT with human P2ry12 and TMEM119 is distinctly visible. Additionally, we have updated the hTMEM119 staining images in Figure 5D. The results from hCD11b staining align with those observed in mouse CD11b staining, notably showing more effective staining in the outer plexiform layer (OPL) microglia cells. The reason for this—whether it pertains to a staining issue, a variance in CD11b expression among microglia cells in the OPL and ganglion layer (GL), or differences in the samples due to varying conditions—is not yet clear and warrants further investigation.

(5) Microglia respond to injury by becoming active and lose their expression of the resting state microglial marker, such as P2ry12, which is used in Fig. 6 for detection of migrated microglia. To confirm that these cells indeed respond to injury like native microglia, one should check for activated microglial markers and induction of pro-inflammatory cytokines in the sodium iodate-injury model.

The reviewer's insights are spot-on. We utilized preserved retinas to extract mRNA, which was then reverse-transcribed to cDNA for conducting qRT-PCR using human-specific primers, as detailed in the updated Table 5. The findings revealed that following retinal pigment epithelium (RPE) injury for 3 days, the transplanted hiPSC-derived microglial cells exhibited an increase in the production of inflammatory cytokines and upregulated genes related to phagocytosis, migration, and adhesion. Conversely, there was a decrease in the expression of microglia-specific signature genes and neurotrophic factors, as demonstrated in Figure 7 suppl.

**Reviewer #1 (Recommendations For The Authors):**
Line 52: "Microglia cell repopulation research suggests that: (1) if no injury or infection occurs, retinal microglia cells can sustain their homeostasis indefinitely" - this statement is too strong or delivers a confusing message; it needs clarification or to be backed up by evidence. Recent single cell RNA sequencing analyses suggest that even under a normal condition, residential microglia do not present as a single homeostatic cell cluster, rather a subpopulation of activated inflammatory microglia are constantly detectable in the normal retina. This is likely because normal retinal neurons can be stressed due to various reasons, such as the temporal accumulation of misfolded proteins, exposed to strong light, or ageing, etc.

We appreciate the comments. We changed the sentence to read, "Microglia cell repopulation research suggests that: (1) retinal resident microglia cells can sustain their population with the local dividing and migration if any perturbations do not exceed the threshold of the recovery speed by local neighbor microglia cells."

Line 83: "we applied an appropriate protocol for culturing human iPSC-derived microglia cells" - it would be more appropriate if the word "appropriate" can be replaced by either "unique" or a phrase like "we adopted a (previously published) protocol...".

Thanks! We changed it to “We modified a previously published protocol to culture human iPSC-derived microglia cells.".

Fig. 1F,G: A method of flow cytometry will provide more comprehensive cell quantification for percentages of positively labeled cells than cell counts under high magnification confocal images.

Thanks for the comments! We agreed with the reviewer. Given the experimental resources available, the quantifications of confocal images did provide a reasonable assessment. We will perform flow cytometry analysis in future experiments.

**Reviewer #2 (Public review):**
Weaknesses:Gene expression analysis of mature microglia cells should be better interpreted and it would be beneficial to compare the iPSC-derived microglia gene set to a human microglial cell line (for example, HMC3) instead of myeloid progenitor cells.The way that the manuscript has been written, unfortunately, is not optimal. I recommend that the entire manuscript be edited and proofread in English. The text contains spelling and grammar mistakes, and the manuscript is inconsistent in several parts. The manuscript should also be revised for a scientific paper format.

We appreciate the reviewer's comments and have taken them into consideration along with similar inquiries from Reviewer 1. Following the suggestions, we conducted a comparison of gene expression profiles between our hiPSC-derived microglia and those from fetal/adult brain microglia, as depicted in the updated Fig.2. Suppl. B, C and D; as well as in the Suppl. table 1 and table 2. The correlation analysis demonstrated that the hiPSC-derived microglia cells closely resemble fetal and adult brain microglia, significantly differing from monocytes and inflammatory monocytes. Additionally, we have revised the manuscript to adhere more closely to the conventional scientific format.

**Reviewer #2 (Recommendations For The Authors):**
Specific suggestions for improvement:- Regarding the characterization of human iPSC-derived microglia, P2RY12 is a general hematopoietic cell marker. One cannot judge the maturity of microglia only by P2RY12 expression (for example, line 261). The expression of more specific markers such as TMEM119 and PROS1 should be studied and discussed.

We are thankful for the reviewer's valuable feedback. In response:

We have removed the term "mature" and clarified that the hiPSC-derived microglia we studied are fully differentiated within single-type cell culture conditions.

We performed a comparative analysis of the gene expression profiles between our hiPSC-derived microglia and microglia from human brains, as illustrated in the updated Fig.2. Suppl. B, C and D. The results affirm that hiPSC-derived microglia closely resemble human fetal and adult microglia.

We noted that the expression of TMEM119 in hiPSC-derived microglia under in vitro single-type cell culture conditions is notably low, as shown in the below A. This suggests that the stimulatory factors in our single-type cell culture might not sufficiently induce TMEM119 expression in microglia. The necessity for a retinal environment or interaction with neuronal and/or other glial cells for TMEM119 expression mirrors the behavior of infiltrating peripheral monocytes in pathological conditions, which initially lack TMEM119 but later differentiate into microglial-like macrophages that express TMEM119, as reported by Ma et al. in Sci Rep (2017).

Additionally, our findings suggest that PROS1 is not uniquely characteristic of microglia but is expressed across a variety of cell types. Within our specific culture conditions, we noted a higher expression of PROS1 in microglial progenitor cells, as shown in Author response image 1B and C.

- In Figure 2, Part E, the names of the genes or pathways in the figure are not clear, and are these genes the set that are the most differentially expressed between iPSCs-derived microglia and MPC? The analysis needs more explanation.

We regret any confusion caused by our previous explanation. To clarify, we compiled a list of microglia-enriched genes from the research conducted by Barres BA Lab (Bennett et al., Proc Natl Acad Sci U S A, 2016) and from our own RNA sequencing data of mouse retinal microglia, identifying a total of 130 genes predominantly expressed in microglia (Suppl. Table 3). We then applied this gene list to analyze our hiPSC-derived microglia RNA sequencing data, resulting in the identification of 71 microglia-specific genes. These 71 genes were subjected to Ingenuity Pathway Analysis (IPA) to visualize the signaling pathways involved. The details of these microglia genes can be found in the updated suppl. table 3.

- Lines 124 to 128 mention that high expression of Stat3, IL1b, and IL6 and their central role in pathway analysis emphasize the efficiency of the maturation protocol. Regarding the fact that Stat3, IL1b, and IL6 are contributors to proinflammatory pathways, it is not convincing that the high expression of these genes in iPSC-derived microglia demonstrates the efficiency of the maturation protocol, given that microglia are not stimulated.

Thanks for the comments! We added the sentences about the comparison results between hiPSC-derived microglia and human brain microglia. We have also replaced the “mature” with “functional.” The sentence reads, “Thus, our method of obtaining differentiated microglia is a reliable method to generate a large number of homogenous functional microglia cells.”

- Statistical analysis is missing for some graphs, for example, figures 1-3 and 5.

We appreciate the comments. We have added the statistical results in the revised version.

- The legend for Figure 3 needs to be rewritten. The graphs or applied assays should be explained in the legend, not the interpretation of the data.

The legend was rewritten.

- There is no Figure 3 in the supplement figures file.

We added Figure 3. Suppl.

- hTMEM119 staining in Figure 5, Part D, is mostly background. Please provide another image.

The images were unclear after on-line converting due to the low number of pixels. We replaced them with new hTMEM119 staining images in Figure 5D.

- In line 176, figure 5I has been forgotten to be mentioned.

Thank you very much! We added 5I.

- Lines 241 to 244 state that more than 50% of the AMD-associated genes are highly expressed in retinal microglia according to Fig. discussion suppl A & B. It is not clear that the gene set that was used for analysis is from a healthy retinal microglia or AMD-related ones. Please explain precisely.

Thank you for your feedback. The gene list we referenced originates from a Genome-Wide Association Study (GWAS) that compared patients with Age-related Macular Degeneration (AMD) to healthy cohorts. We did not directly utilize this list in our experiments but referred to it to underscore the importance of microglia cells in the context of AMD.

Some of the English proofreading and manuscript format comments:Line 805: Iba1 is written in lowercase. Is it human IBA1? It is not consistent with the way it is written in the text (in line 117, for example).

Thank you for pointing out the error. We reformed all Iba1 as “Iba1”. The Iba1 we used here are all from Wako (#019–19741), which labels both mouse and human microglial cells.

Line 814: microglia-enriched gene expression instead of microglia-enrich gene expression

Thank you! We changed it.

Line 345: Starting a sentence with lower case letter.

Thank you! We changed it.

Line 342: Myeloid lineage instead of myeloid cell linage.

Thank you! We changed it.

Line 815: What does FPKM stand for? The abbreviations should be explained.

The FPKM is the abbreviation of Fragments Per Kilobase of transcript per Million mapped reads. We added it in the text.

Line 309: The manuscript has occasionally referred to PLX-5622 without a minus. Please follow a uniform format.

We changed all “PLX5622” to “PLX-5622”.

Lines 327-331: should be rewritten.

The mentioned paragraph was rewritten.

Lines 335-340: should be rewritten.

The mentioned sentence was rewritten.

Line 135: qRT-PCR instead of QPCR," as it is also mentioned in the methods and material. The correction also applies to all the QPCRs in the text.

We changed “QPCR” with “qRT-PCR”

Figure 3: Graph B should be right side of graph AImages description: It is better to have the images description in the left side of the image, for example, figure 5 part B, GL, IPL and OPL

Thanks for the suggestion. We changed the image organization as per the reviewer’s advice.

Lines 258 to 260 in the discussion have also been repeated with the same words in the introduction.

The mentioned paragraph was rewritten.

Lines 327-331 should be rewritten.

The mentioned paragraph was rewritten.

Lines 335-340 should be rewritten.

The mentioned paragraph was rewritten.